# EC-Diffuser: Multi-Object Manipulation via Entity-Centric Behavior Generation

**Carl Qi[1], Dan Haramati[2,3], Tal Daniel[2], Aviv Tamar[2], Amy Zhang[1,4]**
[1] UT Austin, [2] Technion, Israel Institute of Technology, [3] Brown University, [4] Meta AI
`carlq@utexas.edu;dan_haramati@brown.edu`

## Abstract

Object manipulation is a common component of everyday tasks, but learning to manipulate objects from high-dimensional observations presents significant challenges. These challenges are heightened in multi-object environments due to the combinatorial complexity of the state space as well as of the desired behaviors. While recent approaches have utilized large-scale offline data to train models from pixel observations, achieving performance gains through scaling, these methods struggle with compositional generalization in unseen object configurations with constrained network and dataset sizes. To address these issues, we propose a novel behavioral cloning (BC) approach that leverages object-centric representations and an entity-centric Transformer with diffusion-based optimization, enabling efficient learning from offline image data. Our method first decomposes observations into Deep Latent Particles (DLP), which are then processed by our entity-centric Transformer that computes attention at the particle level, simultaneously predicting object dynamics and the agent's actions. Combined with the ability of diffusion models to capture multi-modal behavior distributions, this results in substantial performance improvements in multi-object tasks and, more importantly, enables compositional generalization. We present BC agents capable of zero-shot generalization to performing tasks with novel compositions of objects and goals, including larger numbers of objects than seen during training. We provide video rollouts on our webpage: `https://sites.google.com/view/ec-diffuser`.

## 1 Introduction

Object manipulation is an integral part of our everyday lives. It requires us to reason about multiple objects simultaneously, accounting for their relationships and how they interact. Learning object manipulation is a longstanding challenge, especially when learning from high-dimensional observations such as images. Behavioral Cloning (BC) has shown promising results in learning complex manipulation behaviors from expert demonstrations (Chi et al., 2023; Lee et al., 2024). Recent works (Collaboration et al., 2023; Du et al., 2023; 2024; Zhu et al., 2024) have leveraged vast amount of offline data paired with large models to learn policies from pixel observations. Although scale has proven to be effective in some settings, it is not the most efficient way to deal with problems with combinatorial structure. For instance, despite their impressive generation results, Zhu et al. (2024) require 2000+ GPU hours to train a model on an object manipulation task. In this work, we incorporate object-centric structure in goal-conditioned BC from pixels to produce sample-efficient and generalizing multi-object manipulation policies.

Multi-object environments pose significant challenges for autonomous agents due to the *combinatorial complexity of both the state and goal spaces as well as of the desired behaviors*. Assuming an $n$-object environment with $m$ possible single-object goal configurations, there are $m^n$ total goal configurations and $n!$ orderings of the objects to manipulate in sequence. When learning from offline data, one cannot expect an agent to encounter all possible combinations of objects and desired tasks during training due to either time/compute constraints or lack of such data. We therefore require that our agent generalize to novel compositions of objects and/or tasks it has seen during training, i.e. require *compositional generalization* (Lin et al., 2023).

Reasoning about the different entities becomes increasingly complex when scaling the number of objects in the environment, especially when learning directly from unstructured pixel observations. Object-centric representation learning has shown promise in producing factorized latent representations of images (Locatello et al., 2020; Daniel & Tamar, 2022) and videos (Wu et al., 2023; Daniel & Tamar, 2024) which can be leveraged for learning visual control tasks that involve several objects and possibly facilitate compositionally generalizing behaviors. Prior works have made progress in using these representations for compositional generalization in control (Chang et al., 2023; Zadaianchuk et al., 2021; Haramati et al., 2024). While Haramati et al. (2024) relies on their Transformer-based policy to generalize to unseen configurations in an online reinforcement learning setting, we demonstrate that naïvely taking this approach in the BC setting is insufficient, as the policy fails to capture the multi-modality in behaviors as the number of manipulated object increases. This calls for a method that can better leverage object-centric representations when learning from limited offline demonstrations.

We propose a novel diffusion-based BC method that leverages a Transformer-based diffusion model with an unsupervised object-centric representation named Deep Latent Particles (DLP) (Daniel & Tamar, 2022). We first factorize images into sets of latent entities, referred to as *particles*. We then train a entity-centric Transformer model with diffusion to generate goal-conditioned sequences of particle-states and actions, and use it for Model Predictive Control (MPC). These help deal with the two main challenges in our setting: (1) *Multi-modal Behavior Distributions* – Diffusion models' ability to handle multi-modal distributions aids in capturing the combinatorial nature of multi-object manipulation demonstrations; (2) *Combinatorial State Space* – Our Transformer-based architecture computes attention on the particle level, thus facilitating object-level reasoning which gracefully scales to increasing number of objects, and more importantly, unlocks compositional generalization capabilities. Our Entity-Centric Diffuser (EC-Diffuser) significantly outperforms baselines in manipulation tasks involving more than a single object and exhibits zero-shot generalization to entirely new compositions of objects and goals containing more objects than in the data it was trained on.

## 2 RELATED WORK

**Multi-object Manipulation from Pixels**: Previous work using single-vector representations of image observations for control (Levine et al., 2016; Nair et al., 2018; Hafner et al., 2023; Lee et al., 2024) fall short compared to methods that leverage object-centric representations (Zadaianchuk et al., 2021; Yoon et al., 2023; Haramati et al., 2024; Ferraro et al., 2023; Zhu et al., 2022; Shi et al., 2024) in multi-object environments. Several works have studied compositional generalization in this setting. Collaboration et al. (2023); Du et al. (2023; 2024); Zhu et al. (2024) learn directly from pixel observations with large-scale networks, data and compute and rely on scale for possible generalization and transfer. Other works have proposed approaches that account for the underlying combinatorial structure in object manipulation environments to achieve more systematic compositional generalization (Zhao et al., 2022; Chang et al., 2023; Haramati et al., 2024), requiring significantly less data and compute. We continue this line of work, dealing with the setting of learning from demonstrations and the various distinct challenges it introduces.

**Diffusion Models for Decision-Making**: Diffusion models have been used for decision making in many recent works (Chi et al., 2023; Janner et al., 2022; Ajay et al., 2023; Reuss et al., 2023; Du et al., 2024; Zhu et al., 2024), thanks to their abilities to handle multi-modality and their robustness when scaled to larger datasets and tasks. Diffusion Policies (Chi et al., 2023; Reuss et al., 2023) use diffusion over the actions conditioned on the observations. Diffuser-based approaches (Janner et al., 2021; Zhu et al., 2024) diffuse over both the observations and actions and execute the actions at test time. Finally, Ajay et al. (2023) and Du et al. (2024) train diffusion over the states and train an inverse model to extract the actions. Compared to these works, we build on top of Diffuser (Janner et al., 2022) and generate both object-centric factorized states and actions simultaneously.

## 3 BACKGROUND

In this work, we propose a method that leverages a Transformer-based diffusion model and object-centric representations for learning policies from demonstrations. In the following, we give a brief overview of the different components.

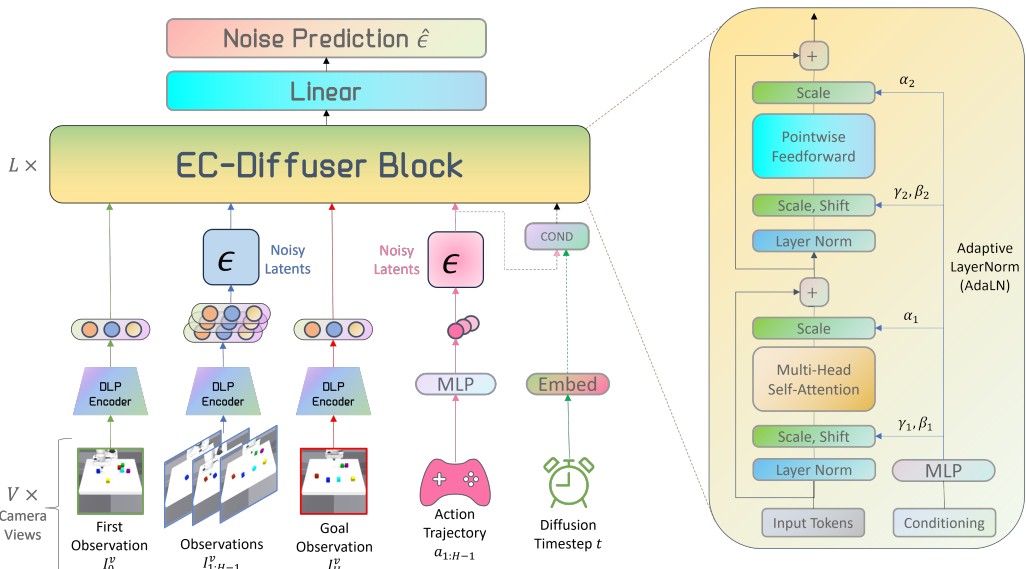

**Figure 1:** EC-Diffuser architecture. Our model learns a conditional latent denoiser that generates sequences of actions and latent states from a trajectory of $H$ image observations across $V$ views $I_{0:H-1}^{0:V-1}$ and actions $a_{1:H-1}$. First, a pre-trained DLPv2 encoder transforms each image into a set of $M$ latent particles $z_{0:H-1}^{0:V-1}$ (where $z$ denotes all $M$ particles for brevity). Projected actions are then added to the latent set as additional particles. A forward-diffusion process adds noise to the actions $a$ and latent states $z$, excluding the first and last (goal) latent states, $z_0^{0:V-1}$ and $z_H^{0:V-1}$, respectively. A Transformer-based conditional denoiser with $L$ blocks predicts the added noise. Each block employs Adaptive LayerNorm (AdaLN) to modulate intermediate variables using a projection of the conditioning variable $t \oplus a_\tau = \text{cat}[t, a_\tau]$, which is a concatenation of the diffusion timestep $t$ and the action $a_\tau$.

**Goal-Conditioned Behavioral Cloning:** Learning-based decision-making algorithms that aim to fit a distribution of offline demonstration data is commonly referred to as Behavioral Cloning (BC). Given a dataset of trajectories $\mathcal{D} = \{(o_t^i, a_t^i)_{t=1}^{T}\}_{i=1}^{N}$ containing environment observations $o \in \mathcal{O}$ and corresponding actions $a \in \mathcal{A}$, the goal of BC is to learn the conditional distribution $\mathbb{P}(a_t|o_t)$ typically referred to as a policy $\pi : \mathcal{O} \to \mathcal{A}$ and parameterized by a neural network. In Goal-Conditioned (GC) BC, the demonstration trajectories are augmented with a goal $g \in \mathcal{G}$ that indicates the task the demonstration was provided for and a goal-conditioned policy $\pi : \mathcal{O} \times \mathcal{G} \to \mathcal{A}$ is learned respectively. The goal can be in the form of e.g. natural language or the last observation in the trajectory $g^i = o_T^i$.

**Diffusion:** In Denoising Diffusion Probabilistic Models (DDPM (Ho et al., 2020)), given a data point $\mathbf{x}_0$ sampled from a real data distribution $q(\mathbf{x}_0)$, a forward diffusion process gradually adds Gaussian noise over $T$ timesteps:

$$q(\mathbf{x}_t|\mathbf{x}_{t-1}) = \mathcal{N}(\mathbf{x}_t; \sqrt{1-\beta_t}\mathbf{x}_{t-1}, \beta_t\mathbf{I}),$$

where $\beta_t$ is a noise schedule. DDPM learns a reverse process to denoise the data:

$$p_\theta(\mathbf{x}_{t-1}|\mathbf{x}_t) = \mathcal{N}(\mathbf{x}_{t-1}; \mu_\theta(\mathbf{x}_t, t), \Sigma_\theta(\mathbf{x}_t, t)).$$

Here, $\Sigma_\theta(\mathbf{x}_t, t)$ represents the learned covariance matrix of the reverse process. In practice, it is often set to a fixed multiple of the identity matrix to simplify the model, i.e., $\Sigma_\theta(\mathbf{x}_t, t) = \sigma_t^2\mathbf{I}$, where $\sigma_t^2$ is a time-dependent scalar. The model is trained to estimate the mean $\mu_\theta$ by minimizing a variational lower bound, which is simplified to a loss $\mathcal{L}$ of predicting the noise $\epsilon$ added during the forward process:

$$\mathcal{L} = \mathbb{E}_{t,\mathbf{x}_0,\epsilon}\left[\|\epsilon - \epsilon_\theta(\mathbf{x}_t, t)\|^2\right],$$

where $\epsilon_\theta$ is the noise prediction network. To incorporate conditional information, the process can be extended to a conditional Diffusion model by conditioning on a variable $\mathbf{c}$, modifying the reverse process to $p_\theta(\mathbf{x}_{t-1}|\mathbf{x}_t, \mathbf{c})$ and the noise prediction network to $\epsilon_\theta(\mathbf{x}_t, t, \mathbf{c})$.

**Deep Latent Particles (DLP):** DLP is an *unsupervised*, object-centric image representation method proposed by Daniel & Tamar (2022) and later enhanced to DLPv2[1] in Daniel & Tamar (2024). DLP

---

[1]Throughout this work, we use DLPv2 but refer to it simply as DLP for brevity.

decomposes an image into a set of $M$ latent foreground particles $\{z^i\}_{i=0}^{M-1}$ and a latent background particle $z_{\text{bg}}$, and is trained in an unsupervised manner as a variational autoencoder (VAE (Kingma & Welling, 2014)), where the objective is reconstructing the original input image from the latent particles. Each foreground particle consists of multiple attributes $z^i = [z_p, z_s, z_d, z_t, z_f]^i \in \mathbb{R}^{(6+n)}$: $z_p \in \mathbb{R}^2$ represents the position in 2D coordinates (i.e., a keypoint); $z_s \in \mathbb{R}^2$ denotes the scale, specifying the height and width of a bounding box around the particle; $z_d \in \mathbb{R}$ approximates local depth when particles are close together; $z_t \in \mathbb{R}$ is a transparency parameter in the range $[0, 1]$; and $z_f \in \mathbb{R}^n$ encodes visual features from a glimpse around the particle, where $n$ is a hyper-parameter determining the latent dimension of the visual features. The background is encoded as $z_{\text{bg}} \in \mathbb{R}^{n_{\text{bg}}}$, with $n_{\text{bg}}$ serving as the latent dimension hyper-parameter for the background encoding. We provide an extended background of DLP in Appendix A.

## 4  METHOD

Our goal is to learn a goal-conditioned offline policy from image inputs for tasks involving multiple objects. A common approach is to develop a model that can generate a sequence of goal-conditioned states and actions, which can then be used for control. The challenge lies in effectively learning such a model from high-dimensional pixel observations. To achieve this, we propose Entity-Centric Diffuser (EC-Diffuser), a diffusion-based policy that leverages DLP, an unsupervised object-centric representation for images, and utilizes a Transformer-based architecture to denoise future states and actions. As DLP is a single-image representation, encoding each image as an unordered set of particles, it lacks correlation between particles across timesteps and views. This property requires a non-trivial design of architecture and optimization objectives. In the following sections, we detail each component of the method. We start by describing the process of encoding images to particles with DLP in Section 4.1, then outline the architecture of our entity-centric Transformer in Section 4.2. Finally, we explain how EC-Diffuser can be applied in goal-conditioned behavior-cloning settings in Section 4.3.

### 4.1  OBJECT-CENTRIC REPRESENTATION WITH DLP

We first extract a compact, object-centric representation from pixel observations. Given image observations of state $s^2$ from $V$ different viewpoints, $(I_s^0, ..., I_s^{V-1})$, we encode each image with a DLPv2 (Daniel & Tamar, 2024) encoder $\phi_{\text{DLP}}$. The resulting representation, as described in Section 3, is a set of $M$ latent particles for each view $v$, denoted by $Z_s^v = \{z_s^{v,i}\}_{i=0}^{M-1}$, where $Z_s^v = \phi_{\text{DLP}}(I_s^v)$. It is important to note that there is no correspondence between particles from different views (i.e. $z_s^{v',i}$ and $z_s^{v'',i}$ can represent different objects), nor is there correspondence between particles from different states (i.e. $z_{s'}^{v,i}$ and $z_{s''}^{v,i}$ can represent different objects). These properties of the DLP representation require a permutation-equivariant policy network architecture, which we describe in the following section.

**Pre-training the DLP:** Following Haramati et al. (2024), we first collect a dataset of image observations to train the object-centric particle representation with DLP. To acquire image data from environments, we employ either a random policy (Haramati et al., 2024) or utilize the demonstration data (Lee et al., 2024). For the BC training stage, we freeze the pre-trained DLP encoder and use it solely for extracting representations. Utilizing these low-dimensional particles instead of high-dimensional images significantly reduces memory strain during the BC stage. We provide detailed information about training the DLP models in Appendix C.3.

### 4.2  ENTITY-CENTRIC TRANSFORMER

Equipped with DLP's object-centric representations, the particles, we aim to construct a conditional generative model. In this work, we adopt a diffusion-based approach (DDPM (Ho et al., 2020)) to learn the denoising of particles and *continuous* actions. However, the unique structure of particles—an *unordered set*—necessitates a permutation-equivariant denoiser. To address this, we design an entity-centric Transformer-based denoiser architecture, termed EC-Diffuser, which processes a sequence of

---

[2]We note that the true state of the environment $s$ is not observed, and the learned DLP representation is trained purely from pixels.

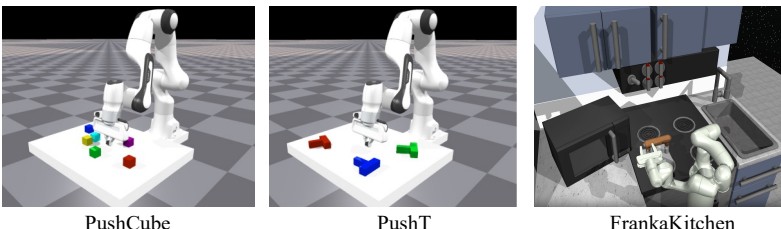

**Figure 2:** Visualization of the simulated multi-object manipulation environments used in this work.

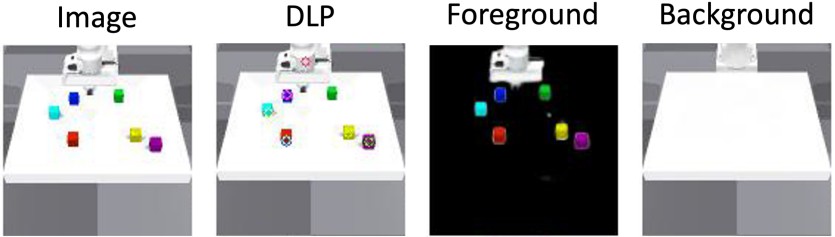

**Figure 3:** Visualization of DLP decomposition in the `PushCube` task. From left to right: the original image, DLP position keypoints overlaid on the original image, reconstructed image from foreground particles, and reconstructed image from the background particle.

observations encoded as particles with DLP, along with corresponding actions. The architecture is illustrated in Figure 1.

Formally, we construct the denoiser inputs as $\{(Z_\tau^v, a_\tau)\}_{v=0,...,V-1,\tau=0,...,H-1}$, where $Z_\tau^v$ denotes the state particles, $a_\tau$ represents the action as a separate token, and $H$ is the generation horizon. Following Haramati et al. (2024), we add the action as an additional particle to the set, as we also generate actions. The EC-Diffuser processes these noised particles and predicts the noise added during the diffusion forward process. Notably, we omit positional embeddings for individual particles, instead incorporating positional information only to differentiate particles from different timesteps and views. Furthermore, the state particles are conditioned on actions and diffusion timesteps via adaptive layer normalization (AdaLN), which has proven beneficial in Transformer-based diffusion models (Peebles & Xie, 2023). Given action $a_\tau$, diffusion timestep $t$, and intermediate variable $z$, AdaLN performs the following modulation in each Transformer block:

$$\alpha_1, \alpha_2, \beta_1, \beta_2, \gamma_1, \gamma_2 = \text{MLP}(\text{cat}[t, a_\tau]),$$

$$z = z + \alpha_1 \cdot \text{Self-Attention}(\gamma_1 \cdot \text{LN}(z) + \beta_1),$$

$$z = z + \alpha_2 \cdot \text{MLP}(\gamma_2 \cdot \text{LN}(z) + \beta_2).$$

### 4.3 ENTITY-CENTRIC DIFFUSER FOR GOAL-CONDITIONED BEHAVIORAL CLONING

We adapt EC-Diffuser to GCBC tasks by incorporating conditioning variables into the diffusion process. Formally, the diffusion process operates over future states and actions: $\mathbf{x}_0 = \{(Z_\tau^v, a_\tau)\}_{v=0,...,V-1,\tau=1,...,H-1}$, with the current state and goal serving as conditional variables: $\mathbf{c}_g = \{(Z_0^v, Z_g^v)\}_{v=0,...,V-1}$. We define the goal as the last timestep in the demonstration trajectory, i.e., $Z_g^v = Z_T^k$, where $T$ is the trajectory length. To train EC-Diffuser, we normalize all input features (DLP's features and actions) to $[-1, 1]$ and employ an $l_1$ loss on both states and actions. One might question the effectiveness of using $l_2$ or $l_1$ losses directly on unordered set inputs. Typically, generating unordered sets like point clouds calls for set-based metrics such as Chamfer distance to compare set similarity. However, in our case, the objective is particle-wise denoising: noise is added independently to each particle, and the denoising process neither imposes nor requires any specific ordering of set elements. Furthermore, we leverage the Transformer's permutation-equivariant structure by omitting positional embeddings within the set of particles. These factors enable the simple $l_1$

loss function to work effectively with diffusion, aligning with previous works that applied diffusion to point clouds (Vahdat et al., 2022; Melas-Kyriazi et al., 2023).

$$\mathcal{L} = \mathbb{E}_{\mathbf{x}_0, t, \mathbf{c}_g, \epsilon} \left[ \| \epsilon - \epsilon_\theta(\mathbf{x}_t, t, \mathbf{c}_g) \|_1 \right].$$

For control purposes, we execute the first action produced by the model in the environment, i.e. $\pi_\theta(\mathbf{x}_t, t, \mathbf{c}_g) = a_0$, and perform MPC-style control by querying the model at every timestep. In practice, we do not directly use the generated latent states for control. However, we empirically found that denoising these latent states is critical, as we discuss later. Notably, the generated latent states serve a valuable purpose for visualization: they can be decoded using the pre-trained DLP decoder to reconstruct images, effectively visualizing the imagined trajectory.

## 5 EXPERIMENTS

The experiments in this work are designed to answer the following questions: (I) How do object-centric approaches compare to unstructured baselines in learning tasks with combinatorial structure? (II) Does object-centric structure facilitate compositionally generalizing behavioral cloning agents? (III) What aspects contribute to performance and compositional generalization?

To study the above, we evaluate our method on 7 goal-conditioned multi-object manipulation tasks across 3 simulated environments and compare against several competitive BC baselines learning from various image representations.

**Environments** A visualization of the environments used in this work is presented in Figure 2, and a visualization of the DLP decomposition for `PushCube` is shown in Figure 3. `PushCube` and `PushT` are both IsaacGym-based (Makoviychuk et al., 2021) tabletop manipulation environments introduced in Haramati et al. (2024), where a Franka Panda arm pushes objects in different colors to a goal configuration specified by an image. In `PushCube` the objects are cubes and the goals are positions, while in `PushT` the objects are T-shaped blocks and the goals are orientations. In `FrankaKitchen`, initially introduced in Gupta et al. (2020), the agent is required to complete a set of 4 out of 7 possible tasks in a kitchen environment. We use the goal-conditioned image-based variant from Lee et al. (2024). Detailed descriptions of these environments as well as the datasets used for training can be found in Appendix B. These tasks all possess a *compositional nature*, requiring the agent to manipulate multiple objects to achieve a desired goal configuration.

**Baselines** We compare EC-Diffuser's performance with the following BC methods: (1) VQ-BeT (Lee et al., 2024): a SOTA, non-diffusion-based method utilizing a Transformer architecture. In our experiments, we find that the ResNet18 (He et al., 2016) used in VQ-BeT fails to achieve good performance in `PushCube` and `PushT`. Consequently, we use a VQ-VAE (Van Den Oord et al., 2017) image representation pre-trained on environment images. (2) Diffuser (Janner et al., 2022): the original Diffuser trained without guidance and takes flattened VQ-VAE image representations as input. (3) EIT+BC – a direct adaptation of the EIT policy from Haramati et al. (2024) to the BC setting, learns from DLP image representations. (4) EC Diffusion Policy: inspired by Chi et al. (2023) and modified for the goal-conditioned setting, learns from DLP image representations. Further descriptions and implementation details of each baseline can be found in Appendix C.2.

For `PushCube` and `PushT`, all results are computed as the mean of 96 randomly initialized configurations. In evaluating `FrankaKitchen`, we adopt the protocol used by VQ-BeT (Lee et al., 2024), sampling 100 goals from the dataset. Standard deviations are calculated across 5 seeds. We provide extended implementation and training details, and report the set of hyper-parameters used in our experiments in Appendix C.

### 5.1 LEARNING FROM DEMONSTRATIONS

In this section we aim to answer the first question – *comparing object-centric approaches to unstructured baselines in learning tasks with combinatorial structure.* Performance metrics for all tasks are reported in Table 1. In `PushCube` and `PushT`, EC-Diffuser significantly outperforms all baselines, with the performance gap widening as the number of objects in the environments increases. Notably, it is the only method that achieves better-than-random performance on `Push3T`, the most challenging task in our suite.

| Env (Metric) | # Obj | VQ-BeT | Diffuser | EIT+BC (DLP) | EC Diffusion Policy (DLP) | EC-Diffuser (DLP) |
|---|---|---|---|---|---|---|
| PushCube | 1 | **0.929 ± 0.032** | 0.367 ± 0.027 | 0.890 ± 0.019 | 0.887 ± 0.031 | **0.948 ± 0.015** |
| | 2 | 0.052 ± 0.010 | 0.013 ± 0.011 | 0.146 ± 0.125 | 0.388 ± 0.106 | **0.917 ± 0.030** |
| (Success Rate ↑) | 3 | 0.006 ± 0.001 | 0.002 ± 0.004 | 0.141 ± 0.164 | 0.668 ± 0.169 | **0.894 ± 0.025** |
| PushT | 1 | 1.227 ± 0.066 | 1.522 ± 0.159 | 0.835 ± 0.081 | 0.493 ± 0.068 | **0.263 ± 0.022** |
| | 2 | 1.520 ± 0.056 | 1.540 ± 0.050 | 1.465 ± 0.034 | 1.214 ± 0.147 | **0.452 ± 0.068** |
| (Avg. Radian Diff. ↓) | 3 | 1.541 ± 0.045 | 1.542 ± 0.045 | 1.526 ± 0.047 | 1.538 ± 0.040 | **0.805 ± 0.256** |
| FrankaKitchen (Goals Reached ↑) | - | 2.384* ± 0.123 | 0.846 ± 0.101 | 2.360 ± 0.088 | **3.046 ± 0.156** | **3.031 ± 0.087** |

**Table 1:** Quantitative performance for different methods in the `PushCube`, `PushT`, and `FrankaKitchen` environments for varying number of objects. Methods are trained for 1000 epochs, and the best performing checkpoints are reported. The best values are in **bold**. *We obtain a lower score than the one reported in the VQ-BeT paper (2.60) using their released codebase, which does not support fine-tuning the ResNet backbone.

Baselines utilizing unstructured representations, such as VQ-BeT and Diffuser, fail entirely when presented with multiple objects. While EIT+BC demonstrates improved performance over these baselines, it struggles to handle more than a single object. This can be attributed to the diverse behaviors present in multi-object manipulation demonstrations, which the deterministic EIT+BC fails to capture. EC Diffusion Policy emerges as the best-performing baseline, differing from our method primarily in that it does not generate states alongside actions. We posit that generating future particle-states serves two purposes: (1) implicitly planning for future object configurations, and (2) acting as an auxiliary objective—similar in spirit to methods such as those proposed by Yarats et al. (2021)—ensuring the model's internal representation is aware of all objects and their attributes.

Our method also outperforms all baselines, including the SOTA method VQ-BeT, on `FrankaKitchen`. In experiments with alternative object-centric representations (Appendix D.2), we found that EC-Diffuser, when paired with Slot Attention (Locatello et al., 2020) achieves state-of-the-art performance with an average of 3.340 goals reached. Intuitively, object-centric representations provide a more useful and structured bias compared to learning from unstructured image representations for tasks such as kitchen object manipulation.

To gain further insight into this performance advantage, we provide DLP decompositions of environment images in Figure 3 and Appendix D.5. These visualizations reveal that particles capture variable aspects of the environment across images—such as the robot, kettle, burner, and parts of hinged doors—while the latent background particle represents the rest. In the context of learning from demonstration images, these captured elements correspond to the *controllable* parts of the environment, which are of primary interest to our agent, thus facilitating better policy learning. A similar notion of disentanglement has proven beneficial in the RL setting (Gmelin et al., 2023). Finally, EC Diffusion Policy also achieves SOTA performance on `FrankaKitchen`, suggesting that this task may not be as challenging as our multi-object tasks.

## 5.2 COMPOSITIONAL GENERALIZATION

The results in Section 5.1 clearly demonstrate that object-centric approaches outperform unstructured methods in multi-object environments when learning from images. As EC-Diffuser is the only method achieving strong performance in manipulating 3 objects, we focus our answer to the second question on whether *our method* can generalize zero-shot to unseen compositions of objects and/or goals. To address this, we consider two generalization settings: (1) `PushCube` – We train our method with 3 objects, their colors randomly chosen at the beginning of each episode from 6 options. We then test it on environments with up to 6 objects. A visualization for the task is shown in Figure 4. (2) `PushT` – We train an agent with 3 objects and 3 goals, and then test on scenarios with up to 4 objects and varying goal compositions. A visualization of this task is shown in Figure 5. For comparison, we present results of the best-performing baseline – EC Diffusion Policy – in these generalization settings. Further details on training these models are provided in Appendix C.1.

We report the quantitative results for `PushCube` generalization in Table 2. Our method significantly outperforms the baseline across all configurations of `PushCube` generalization, maintaining a high success fraction as the number of objects increases. As illustrated in Figure 4(b), which shows the distribution of per-object goal-reaching success, our agent successfully manipulates up to 6 objects to their goals despite being trained on data containing only 3 objects.

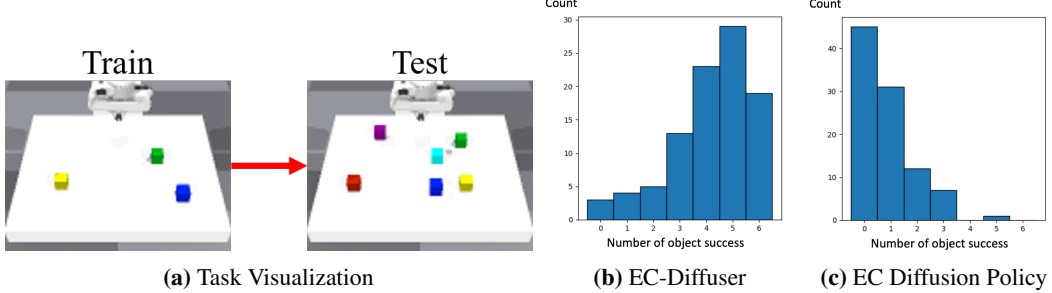

**(a)** Task Visualization        **(b)** EC-Diffuser        **(c)** EC Diffusion Policy

**Figure 4:** `PushCube` Generalization – Agents trained on 3 objects and evaluated on $1 - 6$ objects. See Table 2 for quantitative results. **(a)** Visualization of a `PushCube` generalization task. **(b)**, **(c)** Histograms of per-object goal-reaching success ($n$ out of 6) in the `PushCube` generalization task with 6 objects.

| Number of Objects | 1 | 2 | 3 (training) | 4 | 5 | 6 |
|---|---|---|---|---|---|---|
| VQ-BeT | $0.111 \pm 0.006$ | $0.081 \pm 0.002$ | $0.137 \pm 0.011$ | $0.097 \pm 0.014$ | $0.084 \pm 0.008$ | $0.090 \pm 0.015$ |
| Diffuser | $0.170 \pm 0.036$ | $0.078 \pm 0.018$ | $0.080 \pm 0.012$ | $0.045 \pm 0.015$ | $0.047 \pm 0.004$ | $0.030 \pm 0.011$ |
| EIT+BC | $0.111 \pm 0.006$ | $0.078 \pm 0.018$ | $0.084 \pm 0.012$ | $0.050 \pm 0.014$ | $0.048 \pm 0.012$ | $0.043 \pm 0.006$ |
| EC Diffusion Policy | $0.903 \pm 0.030$ | $0.501 \pm 0.024$ | $0.385 \pm 0.067$ | $0.207 \pm 0.024$ | $0.158 \pm 0.004$ | $0.122 \pm 0.010$ |
| EC-Diffuser (ours) | $\mathbf{0.993 \pm 0.006}$ | $\mathbf{0.981 \pm 0.003}$ | $\mathbf{0.886 \pm 0.051}$ | $\mathbf{0.858 \pm 0.002}$ | $\mathbf{0.767 \pm 0.032}$ | $\mathbf{0.711 \pm 0.070}$ |

**Table 2:** `PushCube` generalization results. The success fraction (number of successful objects / total number of objects) is reported for different numbers of cubes. Agents are trained on 3 objects with colors randomly selected from 6 options. Higher values correspond to better performance The best values are in **bold**.

We additionally report the quantitative results `PushT` generalization in Table 3. These results demonstrate only a slight performance drop (averaging between $0.1$ to $0.26$ radians) when manipulating objects with novel goal compositions for up to $4$ objects. In contrast, EC Diffusion Policy fails to achieve better-than-random performance on these `PushT` generalization tasks. Our approach clearly demonstrates zero-shot generalization capabilities in multi-object manipulation tasks. We showcase the rollouts of our model on the most challenging generalization tasks in Figure 6. We provide additional results on generalization to new objects in Appendix D.4.

### 5.3 ABLATION STUDIES

In this section we aim to answer the third question – *what contributes to the performance of our model?* We ablate our key design choices on the `PushCube` environment. The results, presented in Table 4, report the success rate and success fraction for each task. First, we compare our model to a version that uses VQ-VAE representations as input, treating them as a single particle without any object-centric structure. This unstructured representation results in a significant drop in performance as the number of objects increases, highlighting the importance of the object-centric representation (DLP) for multi-object tasks. Next, we evaluate a model with a similar architecture, but trained without diffusion, where DLP and actions are generated in an auto-regressive manner. For training, we replace the $l_1$ distance with the $l_1$-Chamfer distance for the latent states particles, and use the $l_1$ distance for the actions. This approach fails to learn even in tasks involving a single object, highlighting the critical role of the diffusion process in effectively co-generating DLP and actions. Finally, we compare to a variant of our model that does not generate latent states (as DLP representations). As the number of objects increases, the performance of this ablation rapidly deteriorates, demonstrating that the joint generation of particle states and actions is essential for reasoning about both objects and actions. We provide additional results with alternative object-centric representations in Appendix D.2.

### 5.4 DISCUSSION ON STATE GENERATION

To further explore the generalization ability of our model, we present the generated particle states by decoding them into images using the pre-trained DLP decoder. As shown in Figure 7, our model can produce high-quality future states that were not present in the training data.

Additionally, we analyze how well the model maintains the temporal consistency of particles across frames. To do this, we first label the particles according to the object they represent at each timestep in a trajectory. We then perform a T-test comparing the attention values between particles representing

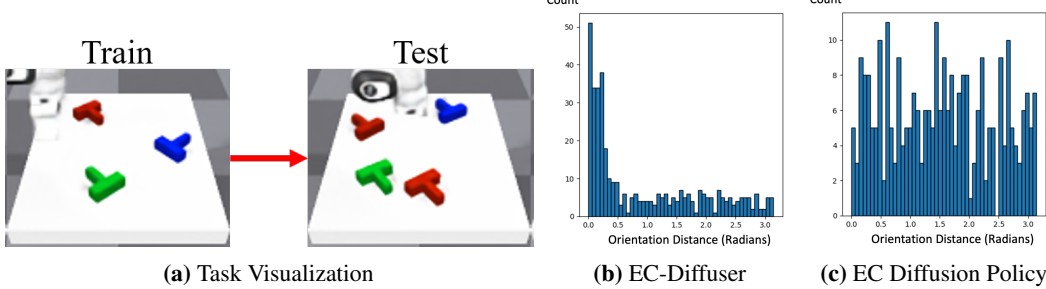

**(a)** Task Visualization        **(b)** EC-Diffuser        **(c)** EC Diffusion Policy

**Figure 5:** `PushT` Generalization – agents trained on 3 objects with 3 goals and are tested in different combinations of objects and goals. See Table 3 for quantitative results. **(a)** Visualization of a generalization task from 3 to 4 objects. **(b)**, **(c)** Histograms of orientation difference (in radians) in the generalization task depicted in **(a)**.

| # of Objects / # of Goals | 3/1 | 3/2 | 3/3 (training) | 4/1 | 4/2 | 4/3 |
|---|---|---|---|---|---|---|
| VQ-BeT | $1.556 \pm 0.082$ | $1.569 \pm 0.053$ | $1.515 \pm 0.036$ | $1.572 \pm 0.086$ | $1.510 \pm 0.005$ | $1.522 \pm 0.043$ |
| Diffuser | $1.597 \pm 0.080$ | $1.541 \pm 0.024$ | $1.545 \pm 0.040$ | $1.632 \pm 0.068$ | $1.541 \pm 0.063$ | $1.527 \pm 0.038$ |
| EIT+BC | $1.515 \pm 0.136$ | $1.507 \pm 0.071$ | $1.501 \pm 0.099$ | $1.554 \pm 0.038$ | $1.594 \pm 0.038$ | $1.524 \pm 0.020$ |
| EC Diffusion Policy | $1.567 \pm 0.027$ | $1.548 \pm 0.117$ | $1.499 \pm 0.013$ | $1.552 \pm 0.071$ | $1.547 \pm 0.040$ | $1.557 \pm 0.020$ |
| EC-Diffuser (ours) | $\mathbf{0.781 \pm 0.160}$ | $\mathbf{0.778 \pm 0.115}$ | $\mathbf{0.817 \pm 0.188}$ | $\mathbf{0.945 \pm 0.050}$ | $\mathbf{0.923 \pm 0.119}$ | $\mathbf{0.948 \pm 0.101}$ |

**Table 3:** `PushT` generalization results, averaged over 96 randomly initialized configurations. The average orientation difference (radians) is reported for various object and goal compositions. The number of colors matches the number of goal objects. When "# of Goals" is less than "# of Objects," multiple objects share the same color and target orientation. Lower values correspond to better performance. The best values are in **bold**.

| Method | 1 Cube | 2 Cubes | 3 Cubes |
|---|---|---|---|
| Unstructured Rep. | $0.829 \pm 0.041$ / $0.829 \pm 0.041$ | $0.089 \pm 0.037$ / $0.221 \pm 0.037$ | $0.008 \pm 0.004$ / $0.128 \pm 0.024$ |
| Without Diffusion | $0.052 \pm 0.024$ / $0.052 \pm 0.024$ | $0.004 \pm 0.005$ / $0.043 \pm 0.010$ | $0.000 \pm 0.000$ / $0.051 \pm 0.010$ |
| Without State Generation | $\mathbf{0.941 \pm 0.020}$ / $\mathbf{0.941 \pm 0.020}$ | $0.423 \pm 0.092$ / $0.496 \pm 0.095$ | $0.529 \pm 0.311$ / $0.715 \pm 0.243$ |
| EC-Diffuser (ours) | $\mathbf{0.948 \pm 0.015}$ / $\mathbf{0.948 \pm 0.015}$ | $\mathbf{0.917 \pm 0.030}$ / $\mathbf{0.948 \pm 0.023}$ | $\mathbf{0.894 \pm 0.025}$ / $\mathbf{0.950 \pm 0.016}$ |

**Table 4:** Quantitative performance for different ablations on `PushCube` tasks. Results are reported as the mean success rate and success fraction (number of object successes / total number of objects) for each task.

the same object over time against the attention values between two random particles. The results show that particles representing the same object over time have significantly higher attention values, with a T-test yielding a p-value of $8.367\mathrm{e}{-6} < 0.05$. This demonstrates that the model implicitly matches objects and enforces object consistency over time, aiding in predicting multi-object dynamics.

# 6   CONCLUSION

In this work, we introduced Entity-Centric Diffuser, a novel diffusion-based behavioral cloning method for multi-object manipulation tasks. By leveraging unsupervised object-centric representations and a Transformer-based architecture, EC-Diffuser effectively addresses the challenges of multi-modal behavior distributions and combinatorial state spaces inherent in multi-object environments. We demonstrated significant improvements over existing baselines in manipulation tasks involving multiple objects, and zero-shot generalization to new object compositions and goals, even when faced with more objects than encountered during training. These results highlight the potential of combining object-centric representations with diffusion models for learning complex, generalizing manipulation policies from limited offline demonstrations.

**Limitations and Future Work**: The performance of EC-Diffuser relies on two core foundations: the quality of the demonstration data and of the object-centric representation. In this work, we utilized DLP, which worked well in our environments. While DLP provides excellent object-centric decomposition of scenes with explicit objects, such as in the `PushCube` and `PushT` environments, it captures slightly different notions of entities in the `FrankaKitchen` environment. Regarding real-world environments, we believe our general approach as well as our proposed algorithm are applicable to real-world object manipulation, and do not see a fundamental limitation in solving tasks similar to the ones in our simulated suite using our method. We provide preliminary EC-Diffuser results on the real-world `Language-Table` dataset (Lynch et al., 2023) in Appendix D.3. That

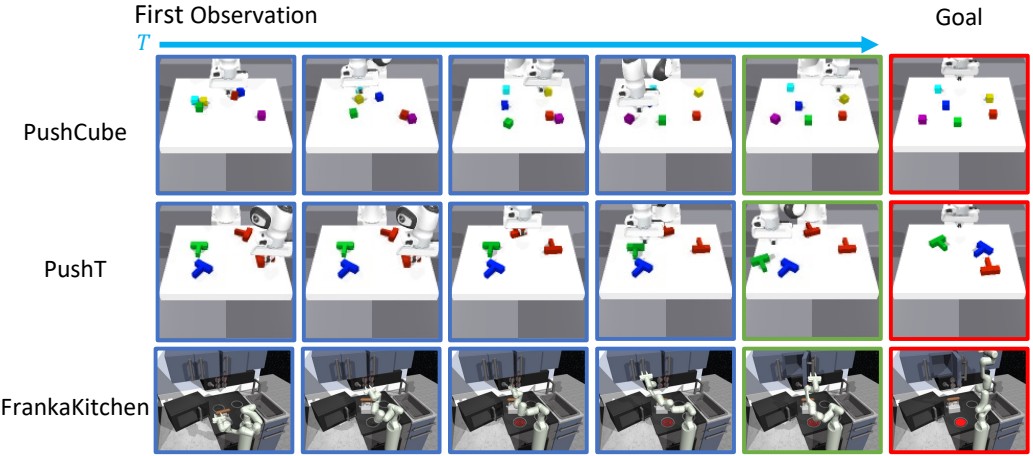

**Figure 6:** Visualization of EC-Diffuser rollouts from each environment. The final observation (highlighted with a green border) demonstrates that our agent successfully completes all tasks.

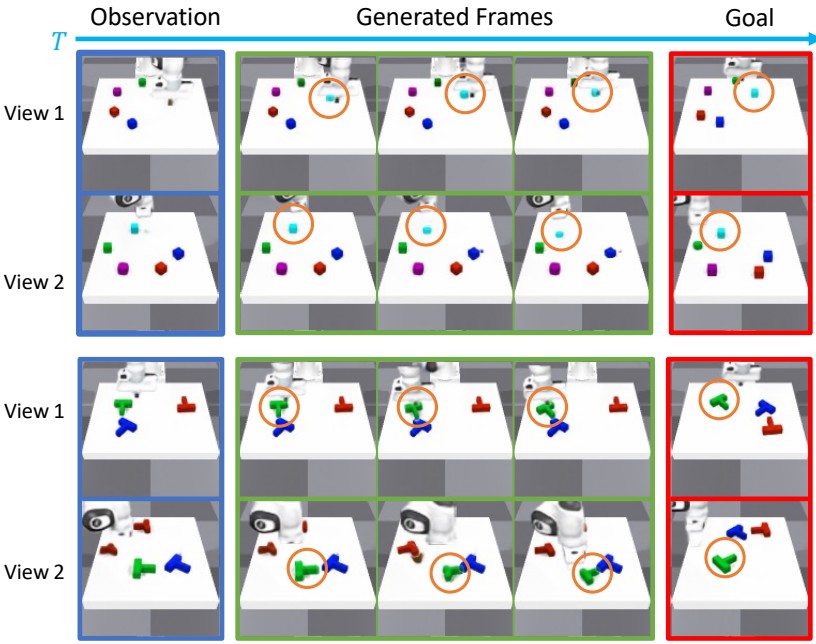

**Figure 7:** Visualization of generated DLP states. The DLP states produced by EC-Diffuser are decoded into images using the pre-trained DLP decoder. (Top) Generated frames for the `PushCube` generalization task: the model is trained with 3 cubes and can generate trajectories involving 5 cubes. (Bottom) Generated frames for the `PushT` generalization task: the model is trained with 3 T-blocks and can generate sequences containing 4. The objects of interest and their respective goals in the generated frames are highlighted with an orange circle.

said, acquiring an unsupervised object-centric representation can be more challenging in real-world scenes due to higher visual complexity, especially in "in-the-wild" environments. The problem of unsupervised object-centric factorization of natural images is far from being solved, and acquiring such representations is an active field of research (Seitzer et al., 2023; Zadaianchuk et al., 2024). Another limitation for real-world application is the long inference time due to the iterative nature of diffusion models. This could be improved by adopting recent approaches focused on reducing the number of iterations required to produce high-quality samples (Song et al., 2020; Karras et al., 2022; Song et al., 2023). Several interesting directions emerge from this work; future research could explore the incorporation of guided diffusion for offline RL, or the application of our approach to planning and world-modeling as well as to new domains such as autonomous driving.

## 7 Reproducibility & Ethics Statement

We provide extended implementation and training details, and report the set of hyper-parameters used in our experiments in Appendix C. We provide the project code to reproduce the experiments at `https://github.com/carl-qi/EC-Diffuser`.

This research was conducted in simulated environments, with no involvement of human subjects or privacy concerns. The datasets used are publicly available, and the work aims to improve the efficiency of robotic object manipulation tasks. We have adhered to ethical research practices and legal standards, and there are no conflicts of interest or external sponsorship influencing this work.

## 8 Acknowledgments

CQ and AZ are supported by NSF 2340651, NSF 2402650, DARPA HR00112490431, and ARO W911NF-24-1-0193. This research was partly funded by the European Union (ERC, Bayes-RL, 101041250). Views and opinions expressed are however those of the author(s) only and do not necessarily reflect those of the European Union or the European Research Council Executive Agency (ERCEA). Neither the European Union nor the granting authority can be held responsible for them.

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

# Appendix

## A   EXTENDED DEEP LATENT PARTICLES (DLP) BACKGROUND

In this section, we present an expanded overview of the Deep Latent Particles (DLP) object-centric representation, as introduced by Daniel & Tamar (2022) and Daniel & Tamar (2024). DLP is an unsupervised, VAE-based model for object-centric image representation. Its core idea is structuring the VAE's latent space as a set of $M$ particles, $z = [z_f, z_p] \in \mathbb{R}^{M \times (n+2)}$. Here, $z_f \in \mathbb{R}^{M \times n}$ encodes visual appearance features, while $z_p \in \mathbb{R}^{M \times 2}$ represents particle positions as $(x, y)$ coordinates in pixel-space, i.e., *keypoints*. Following is a description of the several modifications to the standard VAE framework introduced in DLP.

**Prior:** DLP employs an image-conditioned prior $p(z|x)$, with distinct structures for $z_f$ and $z_p$. $p(z_p|x)$ comprises Gaussians centered on keypoint proposals, generated by applying a CNN to image patches and processed through a spatial-softmax (SSM,Jakab et al. 2018; Finn et al. 2016). The features $z_f$ do not have a special prior neural network, and the standard zero-mean unit Gaussian, $\mathcal{N}(0, I)$, is used.

**Encoder:** A CNN-based encoder maps the input image to means and log-variances for $z_p$ (or offsets from them). For $z_f$, a Spatial Transformer Network (STN) (Jaderberg et al., 2015) encodes features from regions ("glimpses") around each keypoint.

**Decoder:** Each particle is independently decoded to reconstruct its RGBA glimpse patch (where "A" is the alpha channel of each particle). These glimpses are then composited based on their encoded positions to reconstruct the full image.

**Loss:** The entire DLP model is trained end-to-end in an unsupervised manner by maximizing the ELBO, i.e., minimizing the reconstruction loss and the KL-divergence between posterior and prior distributions.

**KL Loss Term:** The posterior keypoints $S_1$ and prior keypoint proposals $S_2$ form unordered sets of Gaussian distributions. As such, the KL term for position latents is replaced with the Chamfer-KL: $d_{CH-KL}(S_1, S_2) = \sum_{z_p \in S_1} \min_{z'_p \in S_2} KL(z_p \| z'_p) + \sum_{z'_p \in S_2} \min_{z_p \in S_1} KL(z_p \| z'_p)$.

**DLPv2:** Daniel & Tamar (2024) extend the original DLP by incorporating additional particle attributes. DLPv2 provides a disentangled latent space structured as a set of $M$ foreground particles: $z = \{(z_p, z_s, z_d, z_t, z_f)_i\}_{i=0}^{M-1} \in \mathbb{R}^{M \times (6+n)}$. Here, $z_p \in \mathbb{R}^2$ and $z_f \in \mathbb{R}^n$ remain unchanged, while new attributes are introduced and described below.

$z_s \in \mathbb{R}^2$: scale, representing the $(x, y)$ dimensions of the particle's bounding box, $z_d \in \mathbb{R}$: approximate depth in pixel space, determining particle overlap order when particles are close, and $z_t \in \mathbb{R}^{[0,1]}$: transparency.

Additionally, DLPv2 introduces a single abstract background particle, always centered in the image and described by $n_{bg}$ latent visual features: $z_{bg} \sim \mathcal{N}(\mu_{bg}, \sigma^2_{bg}) \in \mathbb{R}^{n_{bg}}$. In our work, following Haramati et al. (2024), we discard the background particle from the latent representation after pre-training the DLP. DLPv2 training follows a similar approach to the standard DLP, with modifications to encoding and decoding processes to accommodate the additional attributes.

## B   ENVIRONMENTS AND DATASETS

In this section we give further details about each environment we used in our experiments including how the demonstration datasets were collected and the metrics we use to evaluate performance.

`PushCube`: IsaacGym-based (Makoviychuk et al., 2021) tabletop manipulation environment introduced in Haramati et al. (2024). A Franka Panda robotic arm is required to push cubes in different colors to goal positions specified by images. The agent perceives the environment from two views (front and side) and performs actions in the form of deltas in the end effector coordinates $a = (\Delta x_{ee}, \Delta y_{ee}, \Delta z_{ee})$. Demonstration data for each task (number of objects) was collected by deploying an ECRL (Haramati et al., 2024) state-based agent trained on the corresponding number of objects. We collect 2000 trajectories per task, each containing 30, 50, 100 transitions for 1, 2, 3

objects respectively.

For object-centric image representations, we train a single DLP model on a total of $600,000$ images collected by a random policy from 2 views ($300,000$ transitions) on an environment with 6 cubes in distinct colors. DLP was able to generalize well to images with fewer objects.

To get a wide picture of the goal-reaching performance, we consider the following metrics:

*Success* – A trajectory is considered a success if at the end of it, all $N$ objects are at a threshold distance from their desired goal. The threshold is slightly smaller than the effective radius of a cube.

*Success Fraction* – The fraction of objects that meet the success condition.

*Maximum Object Distance* – The largest distance of an object from its desired goal.

*Average Object Distance* – The average distance of all objects from their desired goal.

`PushT`: IsaacGym-based (Makoviychuk et al., 2021) tabletop manipulation environment introduced in Haramati et al. (2024). A Franka Panda robotic arm is required to push T-shaped blocks to goal orientations specified by images. The object position is not considered part of the task in this setting. The agent perceives the environment from two views (front and side) and performs actions in the form of deltas in the end effector coordinates $a = (\Delta x_{ee}, \Delta y_{ee}, \Delta z_{ee})$. Demonstration data for each task (number of objects) was collected by deploying an ECRL (Haramati et al., 2024) state-based agent trained on the corresponding number of objects. We collect 2000 trajectories per task, each containing 50, 100, 150 transitions for 1, 2, 3 objects respectively.

For object-centric image representations, we train a single DLP model on a total of $600,000$ images collected by a random policy from 2 views ($300,000$ transitions) on an environment with 3 T-blocks in distinct colors. DLP was able to generalize well to images with different numbers of objects.

Since any orientation threshold we choose to define success would be arbitrary, we use the following metric to asses performance:

*Average Orientation Distance* – The average distance of all objects from their desired goal orientation in radians. Since the distance can be considered with respect to both directions of rotation, we always take the smaller of the two. Thus, the largest distance would be $\pi$ and the average of a random policy $\pi/2$ (orientations are uniformly randomly sampled).

`FrankaKitchen`: Initially introduced in Gupta et al. (2020), the agent is required to complete a set of 4 out of 7 possible tasks in a kitchen environment: (1) Turn on bottom burner by switching a knob; (2) Turn on top burner by switching a knob; (3) Turn on a light switch; (4) Open a sliding cabinet door; (5) Open a hinge cabinet door; (6) Open a microwave door; (7) Move a kettle from the bottom to top burner. The action space includes the velocities of the 7 DOF robot joints as well as its right and left gripper, totaling in 9 dimensions. A full documentation can be found in: `https://robotics.farama.org/envs/franka_kitchen/franka_kitchen/`. We use the goal-conditioned image-based variant from Lee et al. (2024), where the environment is perceived from a single view and the goal is specified by the last image in the demonstration trajectory. The demonstration dataset contains 566 human-collected trajectories of the robot completing 4 out of the 7 tasks in varying order with the longest trajectory length being 409 timesteps.

For object-centric image representations, we train a single DLP model on the demonstration data. Performance is measured by the number of goals reached in a trajectory (*Goals Reached*), and the maximum value is 4.

## C  IMPLEMENTATION DETAILS

### C.1  EC-DIFFUSER

We build the Transformer network of EC-Diffuser on top of the Particle Interaction Transformer (PINT) modules from DDLP Daniel & Tamar (2024). We remove the positional embedding for the particles to ensure the Transformer is permutation-equivariant w.r.t the particles. We add different types of positional embedding for views, action particles, and timesteps. We leverage the Diffuser Janner et al. (2022) codebase to train our model: `https://github.com/jannerm/diffuser`. The hyper-parameters we use in training the EC-Diffuser model is shown in Table 5.

We additionally provide details on the model sizes and compute resources used in our experiments in Table 6. For GPUs, we use both NVIDIA RTX A5500 (20GB) and NVIDIA A40 (40GB), though our model training requires only around 8GB of memory. All baseline models have networks of comparable size and are trained on the same hardware.

| Batch size | 32 |
|---|---|
| Learning rate | 8e−5 |
| Diffusion steps | 5, 100 (generalization tasks) |
| Horizon | 3 |
| Number of heads | 8 |
| Number of layers | 6, 12 (generalization tasks) |
| Hidden dimensions | 256, 512 (generalization tasks) |

**Table 5:** Hyper-parameters used for the EC-Diffuser model.

| Model | Parameters | Concurrent GPUs | GPU Hours |
|---|---|---|---|
| EC-Diffuser | 6M | 1 | 12 |
| EC-Diffuser (generalization) | 60M | 4 | 288 |

**Table 6:** Compute resources used for the EC-Diffuser model.

## C.2 BASELINES

**VQ-BeT** (Lee et al., 2024): A SOTA BC method that uses a Transformer-based architecture to predict actions in the quantized latent space of a VQ-VAE. When learning from images, they use a pretrained ResNet18 backbone to acquire an image representation. They experiment with both freezing and finetuning this backbone and report improved performance when finetuning. At the time of writing this paper, code implementing training with finetuning the ResNet was not available. We therefore experimented with either using a frozen ResNet or a VQ-VAE encoder pretrained on environment images as image representations. We report results from the best performing variant in each environment. We use the official code implementation that can be found in: `https://github.com/jayLEE0301/vq_bet_official`.

**Diffuser** (Janner et al., 2022): A diffusion-based decision-making algorithm that we built our method on, thus providing a natural baseline. Diffuser trains a U-Net diffusion model to simultaneously generate entire state-action sequences. Being in the BC setting, we use the variant that uses unguided sampling. As the original paper does not deal with pixel observations, we provide the model with pretrained image representations. We use a VQ-VAE encoder to extract a latent representation of images and flatten it for compatibility with the 1D U-Net architecture. For FrankaKitchen, we use the representation provided by the frozen ResNet18. We use the official code implementation that can be found in: `https://github.com/jannerm/diffuser`.

**EIT+BC**: This method implements a naive adaptation of ECRL (Haramati et al., 2024) to the BC setting by training the Entity Interaction Transformer (EIT) architecture as a BC policy with an $l1$ loss on the actions. It uses the DLP representations of images, as in our method. We use the official code implementation of the EIT that can be found in: `https://github.com/DanHrmti/ECRL`.

**EC Diffusion Policy**: An entity-centric diffusion policy inspired by Chi et al. (2023). It uses the DLP representations of images and has a similar architecture to ours but generates action-only instead of state-action sequences. The difference in the architecture we use for this method is that it uses a encoder-decoder Transformer module. The particles are first encoded with a Transformer encoder with self-attentions. Then, in the decoder, we interleave self and cross attention between the actions and the particle embedding to obtain denoised actions. The hyper-parameters used for this method are described in Table 7. For the implementation of this method we use the same codebase as for our method.

## C.3 PRE-TRAINED REPRESENTATION MODELS

**Data**: For the IsaacGym environments, similarly to Haramati et al. (2024), we collect 600k images from 2 viewpoints by interacting with the environment using a random policy for 300k timesteps. For all methods, we use RGB images at a resolution of $128 \times 128$, i.e., $I \in \mathbb{R}^{128 \times 128 \times 3}$. For FrankaKitchen, we use the offline demonstration dataset collected by Lee et al. (2024) that contains 566 trajectories and around 200k images.

| Batch size | 32 |
|---|---|
| Learning rate | 8e−5 |
| Diffusion steps | 5 |
| Horizon | 5, 16 (FrankaKitchen) |
| Number of heads encoder | 8 |
| Number of layers encoder | 6, 12 (generalization tasks) |
| Number of heads decoder | 8 |
| Number of layers decoder | 6, 12 (generalization tasks) |
| Hidden dimensions | 256, 512 (generalization tasks) |

**Table 7:** Hyper-parameters used for the EC Diffusion Policy model.

**Deep Latent Particles (DLP)** (Daniel & Tamar, 2022): We follow Haramati et al. (2024) and train DLPv2 using the publicly available codebase: `https://github.com/taldatech/ddlp` on the image datasets. Similarly to Haramati et al. (2024), we assign the background particle features a dimension of 1, and discard the background particle for the BC stage. The motivation for this is to limit the background capacity to capture changing parts of the scene such as the objects or the agent. We use the default recommended hyper-parameters and report the data-specific hyper-parameters in Table 8.

| Batch size | 64 |
|---|---|
| Posterior kp $M$ | 24 (IsaacsGym), 40 (FrankaKitchen) |
| Prior kp proposals $L$ | 32 (IsaacsGym), 64 (FrankaKitchen) |
| Reconstruction loss | MSE |
| $\beta_{KL}$ | 0.1 |
| Prior patch size | 16 |
| Glimpse size $S$ | 32 |
| Feature dim $n$ | 4 |
| Background feature dim $n_{\text{bg}}$ | 1 |
| Epochs | 60 (IsaacsGym), 250 (FrankaKitchen) |

**Table 8:** Hyper-parameters used for the Deep Latent Particles (DLP) object-centric model.

**Vector-Quantized Variational AutoEncoder (VQ-VAE)** (Van Den Oord et al., 2017): We follow Haramati et al. (2024) and train VQ-VAE models using their publicly available codebase: `https://github.com/DanHrmti/ECRL`. This aims to provide an unstructured representation in contrast to DLP. We use the default recommended hyper-parameters and report the data-specific hyper-parameters in Table 9.

| Batch size | 16 |
|---|---|
| Learning rate | 2e−4 |
| Reconstruction loss | MSE |
| $\beta_{KL}$ | 0.1 |
| Prior patch size | 16 |
| N embed | 1024 (PushCube), 2048 (PushT) |
| Embed dim | 16 |
| Latent dim | 256 |
| Epochs | 150 |

**Table 9:** Hyper-parameters used for the VQ-VAE model.

# D  ADDITIONAL RESULTS

## D.1  ADDITIONAL METRICS

First, we report the success fraction (i.e., the proportion of objects that meet the success condition) for `PushCube` across all methods in Table 10. This aims to supplement the findings presented

in the main experimental table by offering per-object success data. For EC-Diffuser, the success fraction remains consistent as the number of objects increases, highlighting our method's superior performance in multi-object manipulation. Additionally, we provide more comprehensive results on `PushCube` generalization in Table 11, which includes additional metrics (i.e. Maximum Object Distance, Average Object Distance) from the task.

| Env (Metric) | # Obj | VQ-BeT | Diffuser | EIT+BC (DLP) | EC Diffusion Policy (DLP) | EC-Diffuser (DLP) |
|---|---|---|---|---|---|---|
| `PushCube` | 1 | **0.929 ± 0.032** | 0.367 ± 0.027 | 0.890 ± 0.019 | 0.887 ± 0.031 | **0.948 ± 0.015** |
| | 2 | 0.207 ± 0.020 | 0.083 ± 0.010 | 0.342 ± 0.140 | 0.443 ± 0.086 | **0.948 ± 0.023** |
| (Success Rate ↑) | 3 | 0.097 ± 0.015 | 0.054 ± 0.012 | 0.396 ± 0.239 | 0.807 ± 0.121 | **0.950 ± 0.016** |

**Table 10:** Success fractions for different methods in the `PushCube` environments for varying number of objects. Methods are trained for 1000 epochs, and best performing checkpoints are reported. The best values are in **bold**. The values are computed as the mean over 96 randomly initialized configurations, and standard deviations are computed across 5 seeds.

| Number of Objects | Success Rate ↑ | Success Fraction ↑ | Max Obj Dist ↓ | Avg Obj Dist ↓ |
|---|---|---|---|---|
| 1 | 0.993 ± 0.006 | 0.993 ± 0.01 | 0.011 ± 0.001 | 0.011 ± 0.001 |
| 2 | 0.968 ± 0.010 | 0.981 ± 0.003 | 0.023 ± 0.005 | 0.016 ± 0.002 |
| 3 (training) | 0.833 ± 0.031 | 0.886 ± 0.051 | 0.056 ± 0.020 | 0.030 ± 0.009 |
| 4 | 0.625 ± 0.018 | 0.858 ± 0.002 | 0.114 ± 0.013 | 0.045 ± 0.003 |
| 5 | 0.448 ± 0.057 | 0.767 ± 0.032 | 0.198 ± 0.025 | 0.070 ± 0.011 |
| 6 | 0.240 ± 0.075 | 0.711 ± 0.070 | 0.253 ± 0.056 | 0.089 ± 0.023 |

**Table 11:** Our method's compositional generalization performance on different numbers of cubes in the `PushCube` environment, trained on 3 objects in random colors chosen out of 6 options. The values are computed as the mean over 96 randomly initialized configurations, and standard deviations are calculated across 5 seeds.

## D.2 ADDITIONAL OBJECT-CENTRIC REPRESENTATIONS

We provide experimental results with different types of input to the EC-Diffuser on the `PushCube` and `FrankaKitchen` environments. The results are shown in Table 12 and 13 respectively. All methods use the same architecture and training procedure as EC-Diffuser.

`PushCube`: *Slot Attention* – We train a Slot Attention model (Locatello et al., 2020) to produce a set of latent "slot" vectors. Here, we employ the same slot attention model used in ECRL (Haramati et al., 2024), which produces 10 slots per image. We treat each slot as an entity and pass them into the same EC-Transformer model as used in our method. This variant achieves good results for 1 Cube, but its performance deteriorates quickly as the number of cubes increases, although it remains better than that of the non-object-centric baselines. We attribute this to the fact that, as shown in Figure 15, the slot model occasionally has trouble with individuating nearby objects and represents them in a single slot. *Ground-truth State* – we extract the object location information from the simulator and append a one-hot identifier to distinguish each object. Each (position, one-hot) vector is treated as an entity to be passed into the EC-Transformer. This variant is meant to shed light on the efficacy and generality of our approach by emulating a "perfect" entity-level factorization of images. With these entity-centric set-based state representations EC-Diffuser achieves slightly better performance than using the DLP representation, as expected. *Single View* – This variant only inputs the DLP representation of the front-view image into the EC-Transformer model. We see a drop in performance in this case, highlighting the importance of the ability of EC-Diffuser to effectively leverage multi-view perception in order to mitigate the effect of occlusion and leverage complimenting observational information.

`FrankaKitchen`: We additionally train EC-Diffuser on top of a Slot Attention model trained from the `FrankaKitchen` demonstration data. We use 10 slots, each with a latent dimension of 64. We train the model for 100 epochs on multiple seeds, observe the loss has converged and take the best performing seed. EC-Diffuser with Slot Attention achieves state of the art performance (3.340), surpassing both EC-Diffuser with DLP (achieving 3.031) and VQ-BeT (with a reported performance of 2.60). Based on the slot decompositions in FrankaKitchen (visualized in Figure 16), it is difficult to conclude that its superior performance is due to its ability to capture the objects

| Input Variation | 1 Cube | 2 Cubes | 3 Cubes |
|---|---|---|---|
| Slot Attention | $0.909 \pm 0.021 / 0.909 \pm 0.021$ | $0.364 \pm 0.027 / 0.463 \pm 0.034$ | $0.255 \pm 0.055 / 0.493 \pm 0.061$ |
| Single View | $\mathbf{0.934 \pm 0.015 / 0.934 \pm 0.015}$ | $0.752 \pm 0.054 / 0.828 \pm 0.041$ | $0.685 \pm 0.021 / 0.837 \pm 0.016$ |
| DLP (ours) | $\mathbf{0.948 \pm 0.015 / 0.948 \pm 0.015}$ | $\mathbf{0.917 \pm 0.030 / 0.948 \pm 0.023}$ | $\mathbf{0.894 \pm 0.025 / 0.950 \pm 0.016}$ |
| Ground-truth State | $0.985 \pm 0.013 / 0.985 \pm 0.013$ | $0.963 \pm 0.015 / 0.958 \pm 0.023$ | $0.916 \pm 0.025 / 0.930 \pm 0.017$ |

**Table 12:** Quantitative performance for different input variations on `PushCube` tasks. All methods use the same architecture and training procedure as EC-Diffuser. Results are reported as the mean success rate and success fraction (number of object successes / total number of objects) for each task.

of interest. Nevertheless, these results demonstrate the generality of our method with respect to compatibility with different object-centric representations.

| Env (Metric) | EC-Diffuser with Slots | EC-Diffuser with DLP |
|---|---|---|
| FrankaKitchen (Goals Reached ↑) | $\mathbf{3.340 \pm 0.097}$ | $3.031 \pm 0.087$ |

**Table 13:** Quantitative performance for `FrankaKitchen` of EC-Diffuser with different object-centric inputs.

### D.3 EC-DIFFUSER ON REAL WORLD DATA

We provide preliminary EC-Diffuser results on real world data in this section. Specifically, we use the *real-world* `Language-Table` dataset (Lynch et al., 2023). We subsample 3000 episodes from the real robot dataset, each padded to 45 images, and we randomly select 2700 episodes for training and 300 for validation. We train the DLP model and EC-Diffuser on the training set. We provide a visualization of EC-Diffuser's particle state generation in Figure 8. As shown in the figure, EC-Diffuser can effectively generate high quality rollouts, which shows promise in applying EC-Diffuser to real world problems. We provide DLP decompositions of images from this dataset in Figure 14.

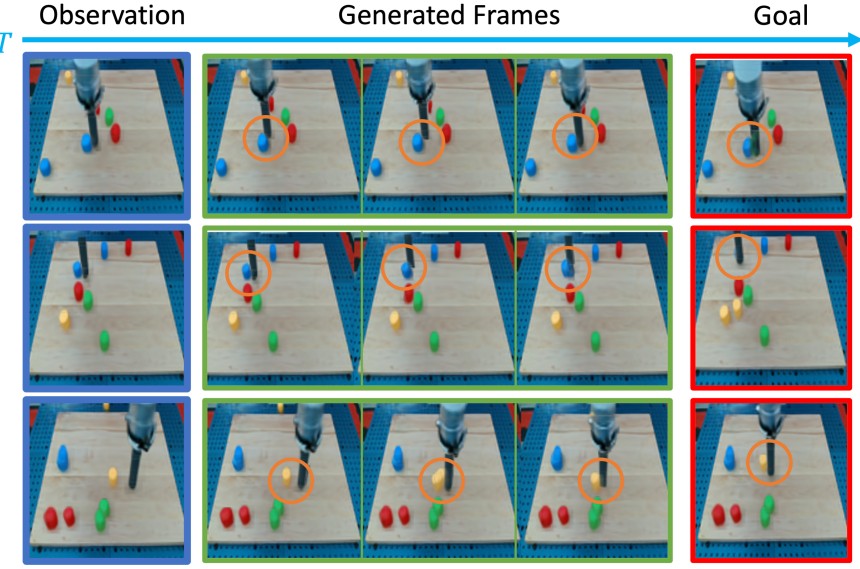

**Figure 8:** Visualization of EC-Diffuser's generation of DLP states of `Language-Table`. The objects of interest and their respective goals in the generated frames are highlighted with an orange circle.

| New Color | Star-Shaped | Rectangular-Shaped | T-Shaped |
|---|---|---|---|

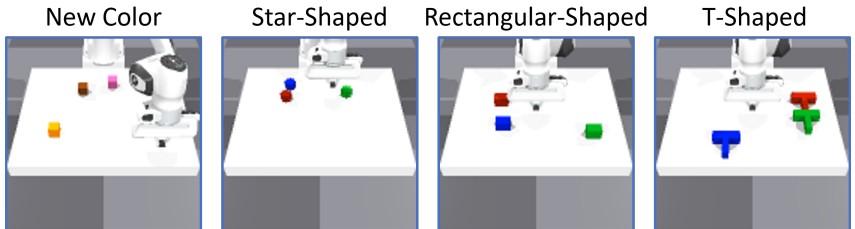

**Figure 9:** Generalization to unseen objects. EC-Diffuser trained on `PushCube` (only cube data in DLP training). From left to right: new colors, star-shaped objects, rectangular-shaped objects and T-shaped objects.

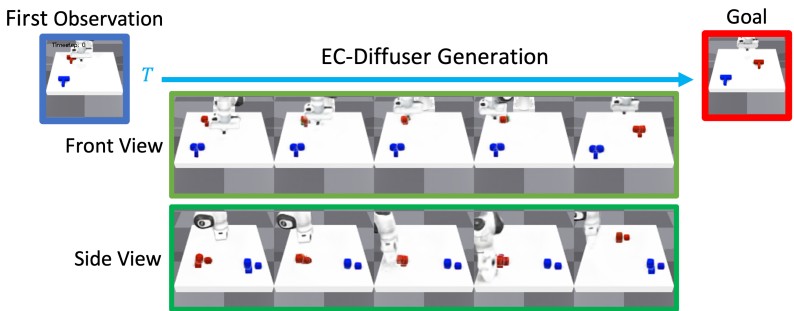

**Figure 10:** We decode the DLP states generated by EC-Diffuser for T-shaped objects. Note that the DLP and EC-Diffuser are both trained only on cubes, but interestingly, the generated shapes are a combination of cubes that resemble the T-shaped blocks.

### D.4 GENERALIZATION TO UNSEEN OBJECTS

We provide additional cases of generalization to unseen shapes and colors as described in Figure 9. We report the performance in Table 14. We see that EC-Diffuser coupled with DLP is able to generalize zero-shot with little to no drop in performance to new colors as well as new shapes (star, rectangular cuboid). When replacing cubes with T-shaped blocks there is a significant drop in performance although success rate is better than random, suggesting some zero-shot generalization capabilities in this case as well. Additionally, we visualize the DLP state generation from the EC-Diffuser (both the DLP encoder and EC-Diffuser trained on `PushCube`) for the T-shaped blocks in Figure 10. We can see that the DLP represents the T-block as a collection of cubes, composing the overall scene from the objects it is trained on. This can be seen as a form of compositional generalization. While we find this is an interesting capability on its own, this generalization is only *visual* and does not always translate to better action or future state generation. EC-Diffuser is still trained for the dynamics of individual cubes and cannot account for dynamics of cubes that are "mended" together to form a T-block.

We see that our policy handles new objects well in cases where they behave similarly in terms of physical dynamics and less when they are significantly different, which is expected.

| Cube Variation | 1 Cube | 2 Cubes | 3 Cubes |
|---|---|---|---|
| New Color | 0.958 | 0.947 | 0.909 |
| Star-Shaped | 0.979 | 0.916 | 0.885 |
| Rectangluar-Shaped | 0.989 | 0.906 | 0.844 |
| T-Shaped | 0.531 | 0.339 | 0.139 |
| Cube (training) | 0.968 | 0.958 | 0.895 |

**Table 14:** Success rate for generalization to different object types on the `PushCube` tasks, computed as the mean over 96 randomly initialized configurations.

## D.5 DLP DECOMPOSITIONS

We provide the visualizations of DLP decomposition by overlaying the particles on top of the original image as well as the showing reconstructions of the foreground and background from the particles. Figure 11 shows the visualization for `PushCube`. Figure 12 shows the visualization for `PushT`. Figure 13 shows the visualization for `FrankaKitchen`. In addition, we train DLP on images from the *real-world* `Language-Table` dataset (Lynch et al., 2023), and provide a visualization of DLP's output in Figure 14. DLP provides accurate decompositions of the scene, indicating it could be paired with EC-Diffuser for real robotic manipulation environments.

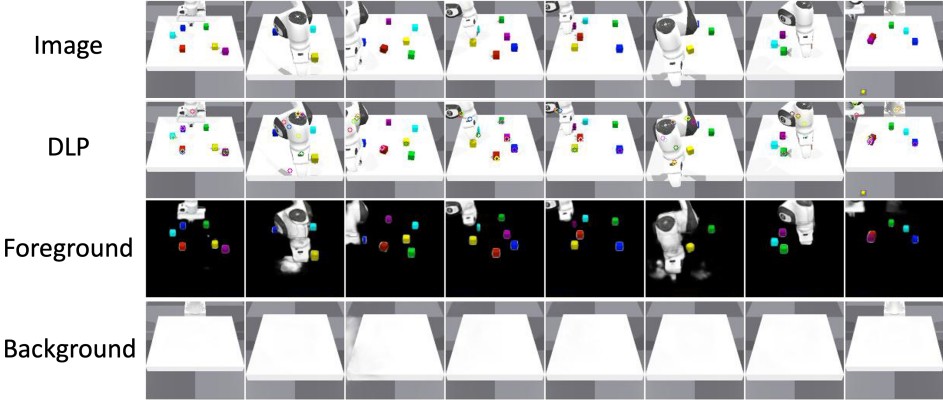

**Figure 11:** Visualization of DLP decomposition of `PushCube`. From top to bottom: original image; DLP overlaid on top of the original image; image reconstruction from the foreground particles; image reconstruction from the background particle.

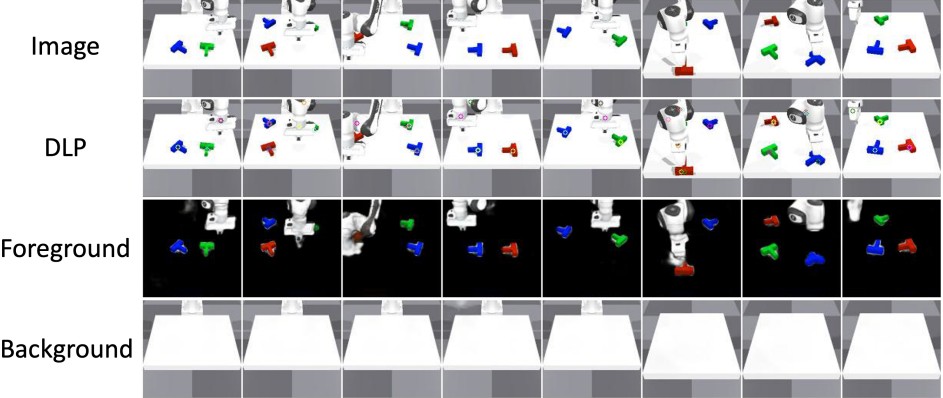

**Figure 12:** Visualization of DLP decomposition of `PushT`. From top to bottom: original image; DLP overlaid on top of the original image; image reconstruction from the foreground particles; image reconstruction from the background particle.

## D.6 SLOT ATTENTION DECOMPOSITIONS

We provide the visualizations of Slot Attention decomposition in this section.

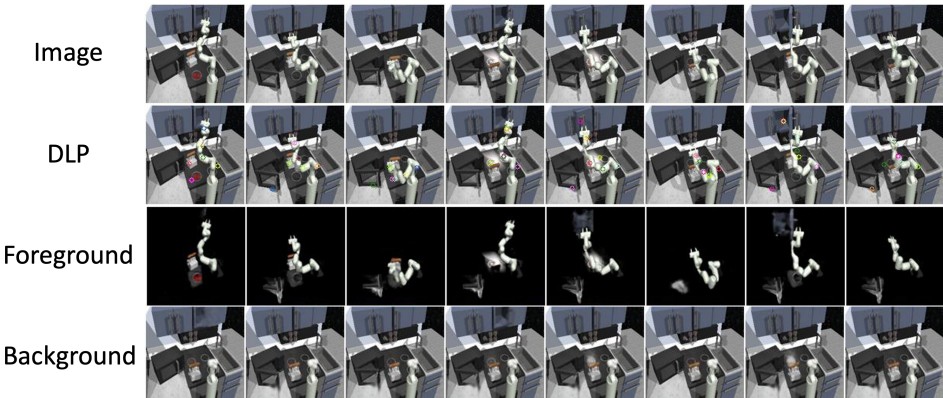

**Figure 13:** Visualization of DLP decomposition of `FrankaKitchen`. From top to bottom: original image; DLP overlaid on top of the original image; image reconstruction from the foreground particles; image reconstruction from the background particle.

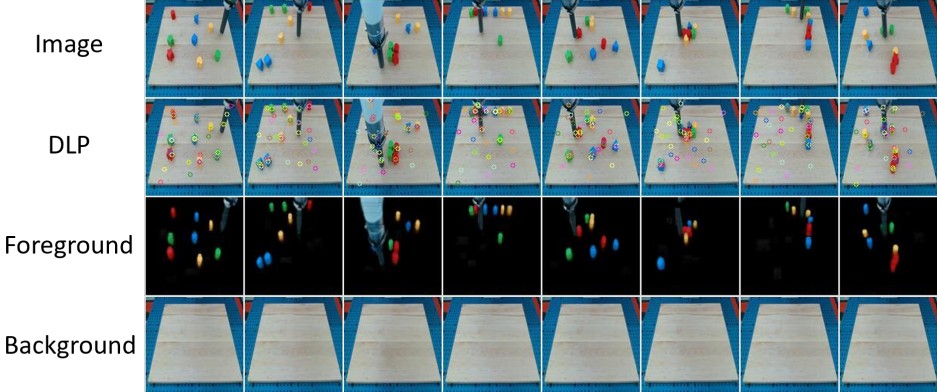

**Figure 14:** Visualization of DLP decomposition of `Language-Table`. From top to bottom: original image; DLP overlaid on top of the original image; image reconstruction from the foreground particles; image reconstruction from the background particle.

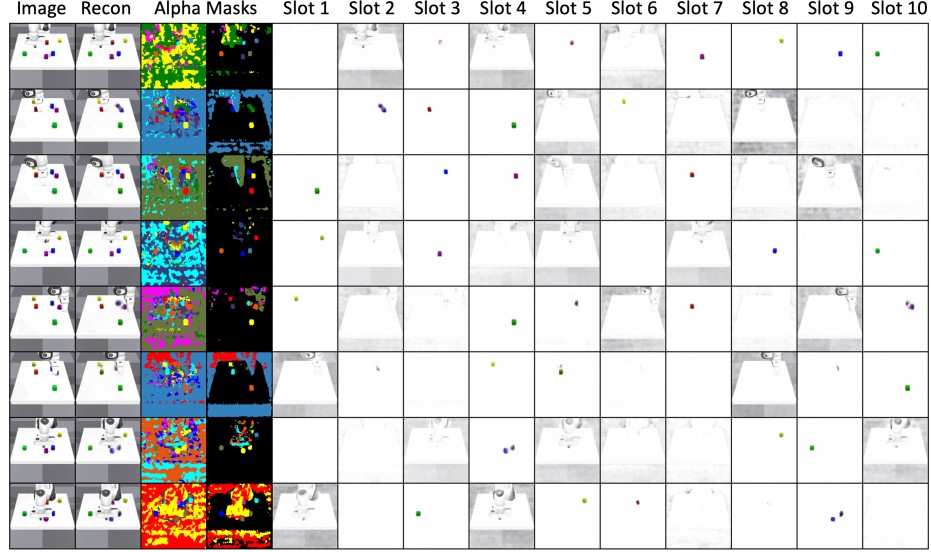

**Figure 15:** Visualization of Slot decomposition of `PushCube`. From left to right: original image; reconstruction from the slots; alpha masks of the slots ; per-slot reconstructions.

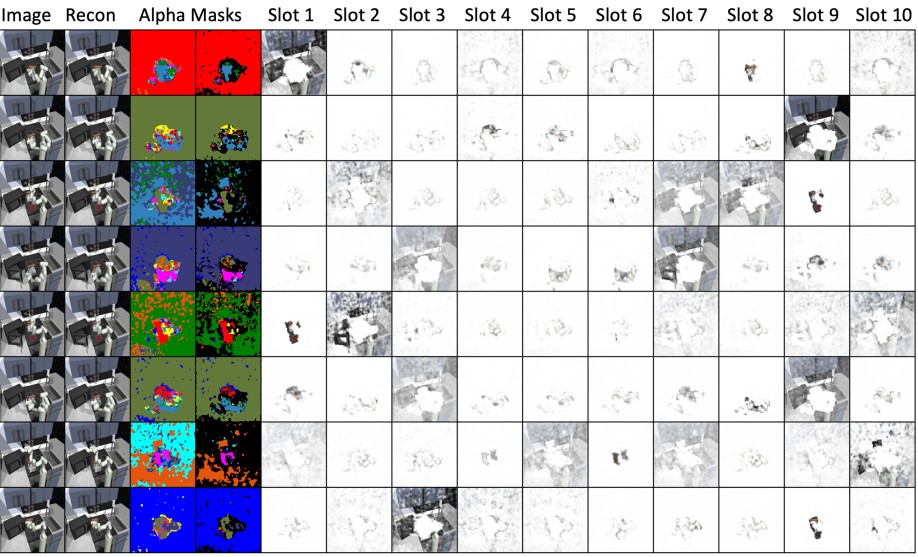

**Figure 16:** Visualization of Slot decomposition of `FrankaKitchen`. From left to right: original image; reconstruction from the slots; alpha masks of the slots ; per-slot reconstructions.

