# OpenReview forum: "EC-Diffuser: Multi-Object Manipulation via Entity-Centric Behavior Generation"
_ICLR.cc/2025/Conference — ICLR 2025 Poster_

### Official Review · Reviewer_WGqn · 2024-10-27

**Soundness:** 3
**Presentation:** 3
**Contribution:** 2
**Rating:** 5
**Confidence:** 4

**Summary:**

The paper presents a method that tackles the problems of behavior cloning and object manipulation. The method contains three main components: an object-centric encoder (DLP encoder), an entity-centric transformer (for temporal information aggregation), and a goal-conditioned diffusion-based decoder (for simultaneous prediction of states and actions). These components were proposed by previous methods, and EC-Diffuser appears to be a combination of several approaches. The results are superior compared to previous work on three simulation benchmarks. Furthermore, it can generalize to scenarios with a higher number of objects (trained on fewer objects, tested on more objects). The ablation studies (Table 5) show the effect of each component, revealing that the DLP encoder and goal-conditioned diffusion parts are the most important ones. Overall, the proposed method appears to be the new state-of-the-art. However, it seems to be a combination of previous methods: DLP (for object encoding), transformer-based sequence modeling (VQ-Bet), and diffusion-based decoding (diffuser). By combining these components, EC-Diffuser gains advantages over previous work through its use of the DLP encoder (better object-centric representations compared to VQ-Bet), improved entity-centric transformer (temporal information aggregation compared to DLPv2, which uses autoregressive prediction), and diffusion decoder (better handling of multimodality and uncertainty compared to VQ-Bet).

**Important note:** This review's technical content and analysis are my original work. Large Language Models were used solely to improve grammar and writing style, without altering any meanings or substantive feedback.

**Strengths:**

* Experiments:
   - The paper evaluates the method on three different benchmarks and provides ablation studies that demonstrate the effect of each component
   - Section 5.4 demonstrates the necessity of state generation, as it helps match objects across different frames. This evaluation approach was both effective and informative
* Performance:
  - The results demonstrate that EC-Diffuser performs exceptionally well on simulation benchmarks and can generalize to different scenarios (as shown in the object-number generalization experiments in Section 5.2)
* Code is available, and video results are presented on the website for better evaluation
* Overall, the paper clearly explains the problem, tasks, and methodology

**Weaknesses:**

* Novelty:
     - As previously stated, EC-Diffuser appears to be a combination of previous work
     - The model incorporates the best components from several predecessors, such as:
          - Object-centric encoder from DLP (while its usage is fair, the results should perhaps not depend so heavily on the DLP encoder; without it, the performance is comparable to VQ-Bet)
          - Transformer encoder from VQ-Bet, which uses a GPT-like transformer encoder on observations encoded by a VQ-VAE
          - Diffusion decoder that predicts both states and actions, similar to Diffuser which also employs this architecture for denoising both state and action
* Real-world evaluation:
    - Can this model be evaluated on real-world datasets such as NuScenes (as VQ-Bet does)?
    - Can EC-Diffuser be adapted for use as GPT-driver?
     - Is it possible to generalize to real-world setups, or would EC-Diffuser be limited to simulation data?
     - What are the necessary changes to make EC-Diffusers work on real-world data?
* Inference runtime cost:
    - As stated in the limitations and future work section, diffusion inference is time-consuming due to its iterative denoising requirement
     - While real-time inference might not be crucial for simulation data, it is essential for tasks such as autonomous driving where real-time decisions are critical

**Questions:**

* As stated in the limitations, the model cannot capture the object-centric notion in scenes from the FrankaKitchen environment. The paper suggests that the DLP encoder could be modified to incorporate additional information, such as ResNet features. Could Slot Attention be used to obtain better object-centric features in FrankaKitchen?
* Could EC-Diffuser be applied to real-world datasets like NuScenes, setting aside its inference cost considerations? (See Weaknesses section as well.)
* Can EC-Diffuser generalize to different time horizons?
    - For instance, if the training set uses a time horizon H, could the model be tested on longer time horizons such as 2H?
    - Are there any architectural considerations that might enable or limit generalization to longer time horizons?
* Can the diffusion model's conditioning mechanism (goal action) be replaced with cross-attention instead of AdaLN?
    - Could the conditioning be improved this way, since any input could be tokenized and fed to the cross-attention layers? The authors' discussion and ideas would be informative.
    - Most state-of-the-art diffusion transformers (DiT) don't use cross-attention but instead use Multimodal DiTs, where both input and condition are tokenized and concatenated (as seen in SD3, SD3.5, etc.). Therefore, the claim about using AdaLN would also be reasonable.
* Minor formatting error: Table 1 and Figure 4 overlap, appearing to be a vertical spacing error.

---

> ### Author Response · Authors · 2024-11-20
>
> We appreciate the reviewer’s acknowledgement of our method’s efficacy and generalization capabilities and of the clarity and reproducibility of our work.
>
> **Novelty**
>
> To the best of our knowledge, we are the first to use diffusion models on top of unsupervised object-centric image representations for *decision-making*, as well as the first to demonstrate such zero-shot compositional generalization capabilities in the behavioral cloning setting.
>
> A major challenge in BC is capturing multi-modal action distributions often present in offline data. This multi-modality is present even in single object manipulation, but increases combinatorially with the number of objects. Diffusion models have demonstrated success in capturing diverse action distributions in the BC setting but without entity-centric structure, they fail catastrophically when scaling the number of objects (as our experiments reveal). One aspect of our work’s novelty is identifying and demonstrating that the above combination works well in the context of multi-object manipulation. To achieve this, we developed a novel Transformer architecture that includes conditioning on actions by incorporating them as a particle state as well as using Adaptive Layer Normalization throughout the denoising diffusion process. In addition, due to the unique structure of the DLP representation, our design choices include equivariance considerations, such as removing positional embeddings.
>
> We would like to respectfully address a few misunderstandings that may be reflected in the reviewer's comments:
>
> *Dependance on DLP* – A key aspect in our method is the use of a *good* object-centric image representation, and one of our goals in this work is to highlight the advantages of leveraging such representations. However, the components of our method that build upon the DLP representation are not inherently tied to the object-centric factorization it provides. To demonstrate this point we conduct experiments showcasing our method’s compatibility with alternative object-centric representations, including Slot-Attention as well as ground-truth state representations. A full analysis and results can be found in Appendix D.2 of the new revision. Our method without *any* object-centric representation nor entity-centric architecture (the unstructured ablation in Table 4), no longer possesses any key aspects of our method and is essentially Diffuser using a Transformer instead of a U-Net architecture. This ablation is comparable to VQ-BeT and thus it is not surprising it achieves equivalent performance.
>
> *Our Transformer Architecture* – We develop a novel Transformer architecture for our purposes, as described above. We do not use nor adapt VQ-BeT’s architecture.
>
> *Choice of Diffuser* – Our method builds upon the Diffuser framework. The best performing baseline is an integration of the core aspects of our proposed method in an action-only generating diffusion model, which is equivalent to Diffusion Policy. It was not clear that the entity-centric Diffuser would perform better than the entity-centric Diffusion Policy in our setting. Part of our contribution is identifying that co-generating states is a key aspect to performance and our analysis of why this is the case (Section 5.1, end of second paragraph and Section 5.4).
>
> **Real-world Evaluation**
>
> We kindly refer the reviewer to our general response regarding *real world applicability*. In addition, we provide answers to the reviewer’s questions:
>
> *Can this model be evaluated on real-world datasets such as NuScenes (as VQ-Bet does)?*
>
> The nuScenes self-driving environment assumes access to an object-centric observation decomposition. Furthermore, it assumes access to object classes and object tracking information (see Figure 11 in the last page of the Appendix of the [VQ-BeT paper](https://arxiv.org/abs/2403.03181 )). In contrast, we do not make these assumptions and propose a method that acquires a representation from images in an unsupervised manner and can leverage it for sequential decision-making without additional assumptions on the structure of the decomposition other than that it is object-centric. While autonomous driving is an interesting domain to incorporate entity-centric structure, the variations in assumptions and the fact it is not an object manipulation environment (which we clearly convey is the focus of our work) are the main reasons we do not include such experiments.
>
> *Can EC-Diffuser be adapted for use as GPT-driver?*
>
> To our understanding, GPT-Driver is a language-model-based approach for motion planning for autonomous driving. We kindly ask the reviewer to clarify the term “adapted for use as GPT-Driver”. Our work uses a Diffusion process, and not language modeling, and we do not rely on LLMs for training or inference.
>
> We do believe EC-Diffuser can be adapted to the autonomous driving domain, and this is an interesting avenue for future work.

---

> > ### Author Response · Authors · 2024-11-20
> >
> > *Is it possible to generalize to real-world setups, or would EC-Diffuser be limited to simulation data? What are the necessary changes to make EC-Diffusers work on real-world data?*
> >
> > While we do not experiment with real robots in the physical world, we believe that our general approach as well as our proposed algorithm are applicable to real-world object manipulation. We do not see a fundamental limitation in solving real world tasks similar to the ones in our simulated suite using our method. That being said, applying our method in the real world would require enhancement to some aspects of our implementation. One would be accelerating diffusion sampling for real-time inference (as the reviewer mentioned) which is not unique to our method but to all diffusion-based decision-making. Solving some tasks might require advancements in unsupervised object-centric image representations to handle the visual complexity of *in-the-wild* environments and more cluttered scenes. Both real-time diffusion and unsupervised object-centric image representation are active lines of work, and advances in those are parallel to our line of work.
> >
> > **Questions**
> >
> > *As stated in the limitations, the model cannot capture the object-centric notion in scenes from the FrankaKitchen environment. The paper suggests that the DLP encoder could be modified to incorporate additional information, such as ResNet features. Could Slot Attention be used to obtain better object-centric features in FrankaKitchen?*
> >
> > DLP is unsupervised, thus what humans define to be distinct objects is not necessarily what DLP (or any other unsupervised object-centric representation) will capture. As explained in the last paragraph of Section 5.1, DLP captures aspects of the environment that change between images. In the case of object manipulation demonstration images, these are normally the agent and objects or parts of objects, which is what we care about when learning visual control policies. The switches might not be captured by the DLP decomposition due to low visual variability (neither their appearance or location change much) but their effects are (burner turns red, light turns on), which still leads to SOTA performance on FrankaKitchen.
> > We experimented with replacing DLP with Slot-Attention as the object-centric representation on FrankaKitchen, results are presented in Appendix D.2, Table 13. In this case, using Slot Attention surpassed DLP and achieved a new SOTA. Judging by the slot decompositions in FrankaKitchen (see Figure 14 in the Appendix), it is difficult to conclude that its superior performance is due to its ability to capture the objects of interest. It is possible that in this case the fact that there are fewer entities (10 slots compared to 40 particles) simplifies training the diffusion model which leads to better performance.
> >
> > *Could EC-Diffuser be applied to real-world datasets like NuScenes, setting aside its inference cost considerations? (See Weaknesses section as well.)*
> >
> > For ease of reading, we repeat our answer to the question “Can this model be evaluated on real-world datasets such as NuScenes (as VQ-Bet does)?” below.
> >
> > The nuScenes self-driving environment assumes access to an object-centric observation decomposition. Furthermore, it assumes access to object classes and object tracking information (see Figure 11 in the last page of the Appendix of the [VQ-BeT paper](https://arxiv.org/abs/2403.03181 )). In contrast, we do not make these assumptions and propose a method that acquires a representation from images in an unsupervised manner and can leverage it for sequential decision-making without additional assumptions on the structure of the decomposition other than that it is object-centric. While autonomous driving is an interesting domain to incorporate entity-centric structure, the variations in assumptions and the fact it is not an object manipulation environment (which we clearly convey is the focus of our work) are the main reasons we do not include such experiments.

---

> > > ### Author Response · Authors · 2024-11-20
> > >
> > > *Generalization to Different Time Horizons*
> > >
> > > This is an interesting question. In general, a training scheme that would allow this type of generalization should be designed, since the model should not be able to handle longer horizons as-is. One aspect limiting generalization to longer horizons is the temporal encoding. This is a learned encoding with a fixed size $H \times F$ where $H$ is the horizon length and $F$ is the entity feature dimension. Other than that, while the Transformer could technically accept longer inputs to denoise, given a start and goal state it learns to produce the $H$ states following the initial state and there is no reason to expect it could predict further timesteps ahead.
> > >
> > > One way to overcome this is using the diffusion model in an *autoregressive* manner: generate a sequence of length $H$ given a start and goal state, then repeat the process by replacing the start state with the final generated state as the condition while leaving the goal unchanged. No architectural change is needed in this case. The limitations in this case are additional inference time and compounding error, but we posit that it should be more accurate for longer horizons than standard autoregressive generation for the same horizon length.
> > >
> > > *Conditioning Mechanism*
> > >
> > > A key aspect of the novelty in our proposed method lies in its architectural design. While it is possible to replace AdaLN with cross-attention to condition on actions and diffusion time-steps, we found AdaLN to be slightly more effective in this context. This observation aligns with the findings in the original [Diffusion Transformer](https://arxiv.org/abs/2212.09748) (see Figures 3 and 5). A similar mechanism was used in [IRASim](https://arxiv.org/abs/2406.14540) to condition video generation of observations on actions.
> > > The effectiveness of AdaLN could arise from several factors. For one, it directly modifies the internal feature representations of the network, offering seamless integration with existing layers. In contrast, cross-attention introduces an additional mechanism that operates at a different abstraction level. Furthermore, for tasks where the relationship between the conditioning signal and the target features involves global modulation – such as the relationship between states and actions (*actions directly influence the state*) – AdaLN may be more effective, as it directly adjusts the feature distribution.
> > >
> > > *Minor formatting error: Table 1 and Figure 4 overlap, appearing to be a vertical spacing error.*
> > >
> > > We thank the reviewer for pointing this out to us. We have fixed the error in the new revision.

---

> > > > ### Comment · Reviewer_WGqn · 2024-11-21
> > > > **response to the comments of authors**
> > > >
> > > > Thank you for your response. First, I want to thank you for conducting new experiments such as Slot Attention and clarifying points. I have several questions/comments which I explain below:
> > > >
> > > > **[Autonomous Driving Case]** Although I agree that adapting your work for autonomous driving would require new assumptions, my main question is: Is EC-Diffuser adaptable to autonomous driving cases? If EC-Diffuser makes these same assumptions and is given the same inputs, would the model be usable in this case? What changes need to be made in EC-Diffuser?
> > > >
> > > > **[Different Time Horizons]** My question is based on video prediction literature, where models are trained to predict K steps into the future. However, inference requires predictions larger than K, such as 2K or 3K. Is there anything limiting this in EC-Diffuser? The authors stated that since the learned encoding is fixed size H×F, that can limit generalization ability, and further, they propose an autoregressive generation procedure. Although it would introduce computational cost and error accumulation, it would allow EC-Diffuser to generate longer timesteps. While it might not be necessary for the datasets used in the work, for future reference, it would be a great experiment showing that a model can generalize to timesteps not seen in the training part.
> > > >
> > > > **[Conditioning]** Thank you for the answer regarding AdaLN. I was thinking of a similar mechanism to adapters in diffusion models, and AdaLN shows promising results as it is used in Diffusion Transformer, Stable Diffusion 3/3.5.
> > > >
> > > > **[Slot Attention]** An experiment with slot attention using 40 slots might give insightful observations since the authors state that fewer entities simplify the training regime and lead the model to better convergence. 40 slots for slot attention will be unnecessary and it probably loses its object-centric notion but being able to where the difference comes from would be informative.
> > > >
> > > > **[Real-world Evaluations]** Although I agree with the authors that real-world evaluation settings would require improvements such as diffusion sampling or better unsupervised object-centric features, it would be good to share some experimental results on real robot data in the physical world.
> > > >
> > > > **[Novelty]** I agree with the authors that this is the first work on unsupervised object-centric image representations for decision-making. However, I still think that the architectural components of EC-Diffusers are taken from previous work (VQ-BeT, DLP, Diffuser). Can the authors clarify how EC-Diffuser differs from each previous work? For DLP, I think there is no difference since it is a visual encoder (Slot attention is also an option). How different is the transformer of EC-Diffuser? For diffusion, is the only difference policy and co-generation of the states?
> > > >
> > > > Again, thank you for the detailed responses.

---

> > > > > ### Author Response · Authors · 2024-11-22
> > > > >
> > > > > We thank the reviewer for their prompt response. We answer the questions in the following:
> > > > >
> > > > > **Autonomous Driving Case**
> > > > >
> > > > > We appreciate the reviewer’s interest in the autonomous driving field. However, this is not the focus of our work, and we are not well-versed in the standard practices of the autonomous driving community. As such, we cannot specify the exact requirements for EC-Diffuser to be applicable to tasks in this domain. We can try to address specific parts of the model given a more detailed description of the setting or task. That said, we believe that given a comparable state representation and assuming the action space is similar to that of robotic manipulation, there are no inherent limitations in our method specific to this application.
> > > > >
> > > > > **Different Time Horizons**
> > > > >
> > > > > Combining diffusion with autoregressive prediction for video generation and world modeling is an emerging area of interest (see [Autoregressive Diffusion Models](https://arxiv.org/abs/2110.02037), [Progressive Autoregressive Video Diffusion Models](https://arxiv.org/abs/2410.08151), [ART-V](https://arxiv.org/abs/2311.18834)), and we share the reviewer’s enthusiasm for this direction. However, due to the MPC-style control employed in our work (we generate a horizon of $H$ and then take the first action), such an approach is not necessary for our current objectives. Adapting this method for model-based RL would likely require incorporating autoregressive prediction, which is an exciting avenue we leave for future work.
> > > > >
> > > > > **Slot Attention**
> > > > >
> > > > > We thank the reviewer for their suggestion. A key limitation of Slot Attention is that each slot encodes information spanning the entire image. This constrains the number of slots that can be effectively used during training, typically to a moderate number (up to 12), as each slot requires significantly more memory compared to the latent particles, which represent small, localized regions of the image. We are not aware of any prior work that successfully trains with a large number of slots, as suggested by the reviewer, nor do we currently have the resources to train such a large model.
> > > > >
> > > > > While fewer entities might simplify training of EC-Diffuser, it does not necessarily result in good down-stream decision-making performance, as we see from the PushCube results. On the contrary, we see that in dynamic scenes with multiple moving objects, having many entities to represent the scene is beneficial for performance and compositional generalization. We believe the performance of Slot Attention compared to DLP on FrankaKitchen is more indicative of the environment/task rather than of the general usefulness of the image representation. Nevertheless, these results do teach us that there is no one-model-fits-all for image representations and it is important to develop modular methods that are compatible with different types of object-centric representations to facilitate future research in the field.
> > > > >
> > > > > **Real-world Evaluations**
> > > > >
> > > > > Given the promising results in simulation and the ability of DLP to decompose real-world images as we demonstrate on the Language-Table dataset (Figure 12, Appendix), we are excited about pursuing follow-up work with real robots, tackling some of the limitations of our approach such as accelerating sampling for real-time inference.

---

> > > > > > ### Author Response · Authors · 2024-11-22
> > > > > >
> > > > > > **Novelty**
> > > > > >
> > > > > > In the following, we clarify how components of EC-Diffuser differ from previous work.
> > > > > >
> > > > > > *VQ-BeT*: We would like to kindly emphasize that our method does not incorporate any components from VQ-BeT, including its Transformer. Our method is entirely different, including inputs and outputs of the Transformer, conditioning signals and mechanisms, training objectives and architectural components.
> > > > > >
> > > > > > The main mechanism of VQ-BeT (and BeT) that allows probabilistic action generation is discretization of actions, which they perform in a phase prior to the training of the Transformer policy (they train a VQ-VAE to reconstruct actions from the demonstration dataset). Once they have discrete actions, they can predict discrete probability distributions over these actions using their Transformer (analogous to next-token probability prediction in language modeling). During inference, they sample from this probability distribution and add an action-correcting offset to account for the information loss due to quantization. This offset is predicted by the same Transformer using a separate prediction head. They use *observation-sequences* for both states and goals, thus requiring a Transformer for handling temporal information.
> > > > > >
> > > > > > We summarize the comparison: (1) Input: we input an initial state, a goal, noisy state-sequences and noisy action-sequences to our Transformer while VQ-BeT inputs state-sequences and goal-sequences. (2) Output: we output denoised continuous state-action-sequences while VQ-BeT outputs a distribution over discrete actions (or action-chunks) and corresponding continuous action offsets. (3) Observation Structure: In our case each temporal observation consists of a set of entities rather than a single vector in VQ-BeT, requiring our Transformer to learn temporal *and* entity-level relations simultaneously. (4) Conditioning: in addition to the initial state and goal, we condition on the diffusion timestep and the noisy action-sequences during the diffusion process using AdaLN. (5) Training Objective: We train a Transformer diffusion model with a sequence-denoising objective while VQ-BeT trains a Transformer to predict both discrete action distributions using a cross-entropy loss and an action offset correction head using an L1 loss on the full action (discrete+offset). (6) Architectural Components: our architecture is a set-to-set equivariant Transformer built from AdaLN-conditioned blocks with multiple learned encoding vectors (horizon timestep, diffusion timestep, action vs. state-entity encoding to handle multi-type inputs) while VQ-BeT’s architecture is a sequence-to-single-vector non-conditional Transformer with two predictor heads.
> > > > > >
> > > > > > *DLP*: DLP serves as the visual encoder in our approach. This encoder can be replaced with other object-centric encoders or even a factorized ground-truth state (as we demonstrate in the additional experiments). The core novelty of our work lies in how to efficiently leverage such factored representations for downstream decision-making (not developing the representation itself), which we rigorously analyze in this study.
> > > > > >
> > > > > > *Diffuser*: our method adapts the Diffuser framework for image-based inputs and utilizing object-centric representations. One major difference lies in the architecture. While the original Diffuser employs a UNet-based architecture, we design a Transformer-based architecture which is tailored to handle object-centric structures and satisfies equivariance requirements. We summarize the comparison: (1) We assume access only to image observations while Diffuser uses ground-truth states. (2) Our architecture is Transformer-based operating on the entity-level while Diffuser’s architecture is UNet-based and operates on single-vector states. (3) The diffusion objective remains the same (L1 loss), and both methods generate states and actions.
> > > > > >
> > > > > > We thank the reviewer again for their engagement, and we are happy to answer any additional questions that the reviewer may have.

---

> > > > > > > ### Comment · Reviewer_WGqn · 2024-11-23
> > > > > > > **Official Comment by reviewer WGqn**
> > > > > > >
> > > > > > > Thank you for your thorough reply. I have several points to address:
> > > > > > >
> > > > > > > **[Slot Attention Memory Requirements]**
> > > > > > > There seems to be a misunderstanding about the memory constraints you've described. The memory complexity of Slot Attention follows O(NKT + KMT), where N represents the number of input pixels, K the number of slots, M the slot dimension, and T the number of time steps. Since this scales linearly with K, not quadratically, increasing slots from 12 to 40 should not cause the severe memory limitations you suggest. The main memory bottleneck comes from the attention operation between pixels and slots (NKT term). Recent work such as "SlotDiffusion: Object-Centric Generative Modeling with Diffusion Models" (NeurIPS 2023) has successfully used 24 slots, demonstrating that training with a larger number of slots is feasible. Therefore, training with 40 slots should be possible given your current setup.
> > > > > > >
> > > > > > > **[Architectural Components & Novelty]**
> > > > > > > While I appreciate the detailed technical comparison you provided, my questions about the architecture were simpler - I wanted to understand EC-Diffuser's capabilities and limitations. It would be more helpful to focus on the novel architectural choices that enable better performance and how these advance the field.
> > > > > > >
> > > > > > > **[Future Directions & Limitations]**
> > > > > > > Regarding the autonomous driving case, stating that "this is not the focus of our work" doesn't address whether EC-Diffuser could be adapted to such scenarios given similar inputs and assumptions. For this and other suggested experiments (increased slot numbers, real-world evaluation), even basic theoretical analysis or proof-of-concept studies could provide valuable insights for the research community. A more constructive discussion might address:
> > > > > > >
> > > > > > > * The key components driving EC-Diffuser's superior performance
> > > > > > > * Potential extensions to new domains and applications
> > > > > > > * Current limitations and proposed solutions
> > > > > > >
> > > > > > > These suggestions aim to enrich the discussion and explore your work's broader impact. I look forward to your thoughts on these points.

---

> > > > > > > > ### Author Response · Authors · 2024-11-25
> > > > > > > >
> > > > > > > > We thank the reviewer for their continued engagement.
> > > > > > > >
> > > > > > > > **Slot Attention Memory Requirements**
> > > > > > > >
> > > > > > > > We trained a Slot Attention model with 40 slots, using the same batch size as DLP. It required significantly more GPU memory than DLP (40G v.s.10G), and training it was slower. We believe that we are on track to evaluate this Slot Attention model on FrankaKitchen by the end of the rebuttal period, during which we will give an update on its performance.
> > > > > > > >
> > > > > > > > **Architectural Components & Novelty**
> > > > > > > >
> > > > > > > > The major aspect of our work that advances the field, as we see it, is identifying the potential of the combination of a diffusion-based model for decision-making with object-centric structure (which to the best of our knowledge, we are the first to do). We provide a clear demonstration of solving challenging multi-object tasks from image observation data and achieve zero-shot compositional generalization.
> > > > > > > >
> > > > > > > > Any architectural choice that we made was to facilitate the above.
> > > > > > > >
> > > > > > > > As we have mentioned in previous responses, constructing a set-to-set permutation-equivariant Transformer architecture, incorporating actions as distinct tokens/entities with appropriate additive encoding, using Adaptive Layer Normalization throughout the denoising diffusion process to more precisely condition on actions, and supervising on both state generation and action prediction with the use of Diffusion, are the key components that drive the superior performance. In our experiments, we have demonstrated that the above design choices are key to improving performance in our comparison with the baselines and the ablations of our method. We are unsure what “EC-Diffuser's capabilities and limitations” the reviewer refers to that were not addressed in the paper and in our responses, and we kindly ask the reviewer for further clarifications and to be more specific in case they feel they require more information.
> > > > > > > >
> > > > > > > > **Future Directions & Limitations**
> > > > > > > >
> > > > > > > > The key components driving EC-Diffuser’s superior performance are highlighted throughout the paper and summarized in the first paragraph of Section 6.
> > > > > > > > Regarding autonomous driving, our simple answer to this question is: *yes*, we believe EC-Diffuser can be adapted to such scenarios given similar inputs and assumptions.
> > > > > > > >
> > > > > > > > For instance, for the nuScene dataset, we can first tokenize the mission (action), ego state, trajectory history, and object information the same way that it was done in VQ-BeT.
> > > > > > > > Then, we can directly input these tokens into EC-Diffuser and denoise the mission token and object tokens, which correspond to denoising the action token as well as the object tokens that captures future positions and one-hot encoded class indicator of each object. This is analogous to using EC-Diffuser to denoise the future actions and DLP states in the object-manipulation domain. The difference is that with the additional assumptions (object classes, tracking) from the driving dataset, EC-Diffuser could be further modified to better leverage these inductive biases.
> > > > > > > >
> > > > > > > > Albeit beyond the scope of this work, we agree that it would be interesting to test our method’s performance benefits in this domain. We have added autonomous driving as an interesting domain to apply our method in future work in the appropriate section of our paper.
> > > > > > > >
> > > > > > > > Regarding current limitations and proposed solutions, we kindly refer the reviewer to the *Limitations and Future Work* portion of Section 6 in the new revision, where we address the points raised by the reviewers with regard to limitations and suggest possible lines of work that could provide solutions to them in the future.
> > > > > > > >
> > > > > > > > We hope our responses have provided more clarity to the reviewer. If the reviewer feels there are still missing details, we kindly ask the reviewer to be more specific regarding the aspects of future works and limitations they want us to discuss. Our impression is that we have addressed all of the suggestions in our responses and incorporated them in the new paper revision.

---

> > > > > > > > > ### Author Response · Authors · 2024-11-27
> > > > > > > > > **Followup Response on Slot Attention Experimental Results**
> > > > > > > > >
> > > > > > > > > We trained EC-Diffuser on a 40-slot Slot Attention model and got slightly better performance than with the 10-slot model, at a larger computational cost. Our hypothesis about the number of slots was not accurate in this case, and we conclude that more experiments are required to understand this specific phenomenon in FrankaKitchen. Nevertheless, the fact that EC-Diffuser outperforms the existing SOTA with both DLP and Slot Attention (both configurations) demonstrates EC-Diffuser’s versatility w.r.t the object-centric representation.

---

> > > > > > > > > > ### Comment · Reviewer_WGqn · 2024-11-30
> > > > > > > > > > **Official Comment by Reviewer**
> > > > > > > > > >
> > > > > > > > > > Thank you for your detailed responses and clarifications.
> > > > > > > > > >
> > > > > > > > > > * Clarifying 40-slot experiment proved that the initial idea of simplifying the experiment resulted better results idea wrong, however, as the authors also stated even with 10 or 40 slots EC-Diffuser achieves state-of-the-art results, which shows slot attention is a strong backbone for extracting object-centric features.
> > > > > > > > > >
> > > > > > > > > > * Regarding EC-Diffuser's potential application to autonomous driving, while the authors' proposed adaptation seems theoretically feasible, I have reservations about its practical performance. Autonomous driving presents significantly more complex challenges than robotics tasks, and the features extracted by slot attention may not provide sufficiently rich information for the transformer component. I agree that this might be beyond the scope of the network because this work focuses robotics task but seeing it can be applied to other domains would be good direction for the future research.
> > > > > > > > > >
> > > > > > > > > > * While EC-Diffuser innovates by being the first to combine diffusion-based decision-making with object-centric representations, its individual components largely build upon existing techniques:
> > > > > > > > > >
> > > > > > > > > >    1. The permutation-equivariant Transformer architecture and AdaLN, though well-implemented, are established techniques in the field
> > > > > > > > > >    2. The co-generation of states and actions builds directly on Diffuser's approach
> > > > > > > > > >    3. The main performance gains seem to come from the combination of these known components rather than novel architectural innovations
> > > > > > > > > >
> > > > > > > > > > The paper's primary contribution appears to be the empirical demonstration that this specific combination of techniques works well for multi-object manipulation. While valuable, this is more of an engineering achievement than a fundamental advance in the field.
> > > > > > > > > >
> > > > > > > > > > I maintain my score of 5 but will not oppose the paper's acceptance based on other reviewers' comments and ratings. I appreciate the authors' thorough responses and clarifications throughout this discussion.

---

### Official Review · Reviewer_3tkq · 2024-11-01

**Soundness:** 3
**Presentation:** 4
**Contribution:** 3
**Rating:** 8
**Confidence:** 4

**Summary:**

The authors propose EC-Diffuser, a diffusion-based behavioral cloning method designed to enhance multi-object goal-conditioned manipulation tasks from high-dimensional pixel observations. This approach leverages a pre-trained object-centric encoder to map each observation into an unstructured set of latent vectors, where each vector corresponds to a specific object or the background within the observation. The diffusion model is trained to denoise these latent trajectories paired with corresponding actions, using the initial and goal representations as conditional information. To address the unordered nature of these latent representations, the authors employ a Transformer-based diffusion model due to its permutation equivariance.

The authors validate their approach through experiments on several simulated multi-object manipulation environments, demonstrating that it outperforms baseline methods. They also provide ablation studies to highlight the contributions of different components in their method.

**Strengths:**

* __Writing.__ The paper is well-written and easy to follow. The motivation behind addressing the challenges in multi-object manipulation from high-dimensional pixel observations is clearly articulated.

* __Novelty.__ The authors present a novel integration of existing approaches by combining object-centric representations (Deep Latent Particles) with a diffusion-based behavioral cloning method. The use of an entity-centric Transformer to handle the unordered nature of latent representations is a noteworthy contribution.

* __Positioning.__ The paper is well-situated within the existing body of research, providing a comprehensive review of related work and clearly distinguishing its contributions from prior methods.

* __Experimental design.__ The authors conduct thorough experiments across several simulated multi-object manipulation environments. They adapt and compare against multiple competitive baselines, ensuring a fair and rigorous evaluation of their method. They also thoroughly compare their approach with both structured and unstructured latent representation-based methods, providing a comprehensive evaluation.

* __Performance.__ Experimental results demonstrate that the proposed approach consistently outperforms baseline methods.

**Weaknesses:**

* __Alternative object-centric encoders.__ While the authors utilize DLPv2 as a powerful object-centric encoder, the paper would benefit from experimenting with other similar approaches. Since the DLP module appears to be easily replaceable with other object-centric encoders, exploring alternatives could provide insights into the generality and robustness of the proposed method. This would also demonstrate whether the observed benefits are specific to DLPv2 or applicable across different encoding techniques.

* __More generalization experiments.__ Although the authors present a wide range of experiments showcasing the advantages of their approach, additional generalization studies could strengthen the paper. Specifically, it would be interesting to see experiments involving novel object shapes, textures, or views where the use of object-centric latent representations might not be as beneficial at test time compared to regular unstructured latents.

Minor Presentation Issues:

* There is a rendering issue with Figure 5, where the figure overlaps with the text.

* In Tables 2 and 3, the authors use __bold text__ to indicate the training setup. It might be clearer to mention this in parentheses in the column header. This is a minor stylistic suggestion but could improve the overall presentation.

**Questions:**

* __Table 1 results interpretation.__ In Table 1, regarding the performance of the __EC Diffusion Policy__ on the __PushCube__ environment, there is an unusual pattern where the success rate decreases when moving from one to two objects but then increases from two to three objects. In contrast, all other methods exhibit a consistent decrease in performance as the number of objects increases. Could the authors provide an explanation for this anomaly in their results?

* __Real-world limitations.__ It appears that the object-centric encoder is the main bottleneck when it comes to extending this model to real-world scenarios. Is this assessment correct? If so, do the authors believe that object-centric approaches are preferable for real-world applications, and how might they address the challenges posed by the encoder in practical implementations?

---

> ### Author Response · Authors · 2024-11-20
>
> We thank the reviewer for acknowledging the novelty and contribution of our work and for their genuine interest and attention to detail, providing insightful suggestions for strengthening our paper.
>
> **Alternative Object-centric Encoders**
>
> To the reviewer’s suggestion, we have conducted experiments with alternative object-centric representations, including Slot Attention as well as ground-truth state representations. A full analysis and results can be found in Appendix D.2 of the new revision. In addition, we summarize our findings here for the reviewer’s convenience:
>
> *Ground-truth State* - To shed light on the generality of our approach, we experiment on the PushCube environments using object-centric set-based state representations by extracting the object location information from the simulator and appending a one-hot identifier to distinguish each object (a similar baseline was considered in [ECRL](https://arxiv.org/abs/2404.01220)). This representation is meant to emulate a “perfect” entity-level factorization of images. Our experimental results show that using this representation, the EC-Diffuser achieves slightly better performance than using the DLP representation, as expected. We believe that these results highlight two important aspects: (1) The benefit of entity-centric structure, which motivates such factorizations of images. (2) EC-Diffuser’s ability to handle different types of object-centric representations (the state-based [position, 1-hot] is very different from the DLP representation).
>
> *Slot Attention* - We replace the DLP encoder with a Slot Attention encoder and experiment on the PushCube environments as well as FrankaKitchen. Performance on PushCube using slot-attention-based representations is better than non-object-centric baselines but worse than learning using DLP representations. On the FrankaKitchen environment, EC-Diffuser using Slot Attention surpases DLP, achieving a new SOTA of $3.340$ compared to the previous $3.031$ obtained by EC-Diffuser using DLP. Judging by the slot decompositions of the different environments (see Figure 13 and 14 in the Appendix of the new revision), it seems that Slot Attention occasionally has trouble with individuating nearby objects and represents them in a single slot. We believe this is a major factor in the performance drop in PushCube with multiple cubes. In FrankaKitchen this does not affect performance, we believe this is due to the fact that most objects in the scene are static (and usually allocated together in the same slot, and the robot in another slot, leading to a foreground-background decomposition) thus not necessarily requiring an accurate decomposition for good downstream control. We emphasize that the object-level state complexity in PushCube and PushT is much higher than in FrankaKitchen, making the former significantly more challenging.
>
> In conclusion, these experiments demonstrate the generality of our method with respect to compatibility with different object-centric representations. We further motivate the specific choice of DLP in our general response.
>
> **More Generalization Experiments**
>
> To the reviewer’s suggestion, we evaluate our PushCube generalization policy (trained on 3 cubes in different colors chosen from 6 options) on the following:
> - Replacing cubes with star-shaped objects
> - Replacing cubes with rectangular cuboids
> - Replacing cubes with T-shaped blocks
> - Introducing unseen cube colors
>
> The results can be found in Appendix D.3, Table 14. We see that EC-Diffuser coupled with DLP is able to generalize zero-shot with little to no drop in performance to new colors as well as new shapes (star, rectangular cuboid). When replacing cubes with T-shaped blocks there is a significant drop in performance although success rate is better than random, suggesting some zero-shot generalization capabilities in this case as well. We see that our policy handles new objects well in cases where they behave similarly in terms of physical dynamics and less when they are significantly different, which is expected.
>
> We did not experiment with new camera views but we do not expect the policy to map observations to the correct actions due to the frame-of-reference shift, requiring re-grounding of observations to actions. Policy adaptation via fine-tuning techniques could help deal with novel views and possibly with novel objects with significantly different dynamics.
>
> **Presentation Suggestions**
>
> We thank the reviewer for their suggestions. We have fixed the rendering issue with Figure 5 and have indicated the training setup in Tables 2 and 3 in the column header instead of using bold text.

---

> > ### Author Response · Authors · 2024-11-20
> >
> > **Questions**
> >
> > *Table 1 Results Interpretation*
> >
> > This is indeed an anomaly; however, we reran the same experiment multiple times during the rebuttal period and consistently observed this trend. A common failure mode we observe in that case is that the policy occasionally ignores the colors of the cubes, manipulating the cubes to the correct locations but not matching the goal colors. We see a similar trend for the ablation of our method without generating states. Since in both cases the diffusion model does not generate states, we attribute this behavior to the benefits of generating states: ensuring the model’s internal representation is aware of all objects and their attributes (see Section 5.1, end of second paragraph), in this case color. We hypothesize that the anomaly is due to the fact this phenomenon is more prevalent with fewer objects and lower color variability.
> >
> > *Real-world Limitations*
> >
> > In the following, we provide multiple perspectives for real-world applicability.
> > *Similar environments and tasks, but in the real world as opposed to simulation*: we believe our method can be applied to such environments using DLP. We provide DLP decompositions of real images from the [Language-Table](https://interactive-language.github.io/) dataset (an environment similar to PushCube) in Appendix D.4, Figure 12 of the new revision. The main limitation we see currently in this setting is real-time decision-making using diffusion models, which is not unique to our approach but shared among all diffusion-based decision-making. We discuss this and possible solutions, such as works dedicated to accelerating diffusion, in the “Limitations and Future Work” portion of Section 6.
> > *In-the-wild and open-world environments*: we agree that acquiring an unsupervised object-centric representation can be more challenging due to the visual complexity of such environments. Although DLP has shown some success in factorizing real-world images (see results on the Traffic dataset in the [DDLP](https://arxiv.org/abs/2306.05957) paper), the problem of unsupervised object-centric factorization of natural images is far from being solved. Acquiring such representations is an active field of research (e.g. [“Bridging the Gap to Real-World Object-Centric Learning”, Seitzer et al.](https://arxiv.org/abs/2209.14860)) and progress in this direction is orthogonal to our line of work. Our work demonstrates that object-centric approaches incorporating object-level factorization of images and corresponding object-centric architectures have great potential in solving complex multi-object tasks and facilitating compositional generalization. We believe these findings point to a fundamental aspect that applies to real-world environments as well: perceiving the world in the form of sets of entities and how they interact is a useful structural bias in decision-making problems, including (but not limited to) robotic object manipulation.
> >
> > We have added a discussion on this subject under “Limitations and Future Work” in Section 6 in the new revision.

---

> > > ### Comment · Reviewer_3tkq · 2024-11-24
> > >
> > > I would like to thank the authors for the detailed and thoughtful reply, as well as the additional experiments and discussions provided. The extra analyses and clarifications have addressed my concerns effectively. I will maintain my current rating for this submission.

---

### Official Review · Reviewer_1WDY · 2024-11-03

**Soundness:** 3
**Presentation:** 3
**Contribution:** 3
**Rating:** 6
**Confidence:** 2

**Summary:**

This paper propose a diffusion-based behavioral cloning method. It uses the object-centric representations instead of pixel-level representation. Deep Latent Particle is used to get representation, followed by an entity-centric Transformer at the particle level to predict action sequences.

**Strengths:**

- This idea of predicting object manipulation actions through a diffusion-based architecture is interesting. By incorporating object-level information, the proposed method achieves a performance improvement over baseline methods.

- This paper also compare with the state-of-the art non-diffusion baseline, i.e., VQ-BeT, demonstrating the effectiveness of the proposed method.

**Weaknesses:**

- The method depends on the capabilities of the image representation algorithm DLP, and experiments are conducted only in synthetic environments. It is unclear if it will perform well in more complex settings. For example, the method is only compared with one non-diffusion baseline VQ-BeT, which is preformed extremely bad on PushCube and PushT (2 out of 3 test environments used in this paper). I wonder why it is only tested on these three environments. Is it possible to also test the model on other environments that baselines achieve very good results, such as Multimodal Ant, BlockPush, UR3 BlockPush, or even nuScenes self-driving, to better demonstrate the advantage of the proposed method?

- What are the advantages of using a diffusion model over other transformer-based methods? Could you provide an intuitive explanation for why it is suitable for this task?

**Questions:**

- What's the motivation of using diffusion for this task? What are the advantages of using a diffusion model over other transformer-based methods?

- Is it possible to also test the model on other environments, such as Multimodal Ant, BlockPush, UR3 BlockPush, or even nuScenes self-driving, to better demonstrate the advantage of the proposed method?

---

> ### Author Response · Authors · 2024-11-20
>
> We thank the reviewer for their acknowledgment of the efficacy of our method compared to competitive baselines and for their questions and interest in our work.
>
> **Dependance on Object-centric Image Representation**
>
> Our method strongly depends on a *good* object-centric image representation, and one of our goals in this work is to highlight the advantages of leveraging such representations. However, the components of our method that build upon the DLP representation are not inherently tied to the object-centric factorization it provides.
> In principle, DLP can be replaced with other object-centric representations of images such as slot-based representations. We refer the reviewer to the Slot Attention experiment in Appendix D.2, Table 12 and 13 of the new revision, where we demonstrate the performance of our method coupled with Slot-Attention instead of DLP.
>
>
> The simplest way to define a *good* object-centric representation is one that effectively facilitates downstream decision-making. As our experimental results demonstrate, DLP not only fulfills this criterion but also enables zero-shot generalization in certain scenarios. We elaborate on why we believe DLP is better suited than slot-based models for the types of tasks we address in the general response under *Choice of Object-centric Representation*.
>
> **Simulated Experimental Environments**
>
> We would kindly like to point out that the simulated environments we use in our work are either equally or more challenging than any simulated environment baseline methods have tested on in terms of multi-object complexity, goal space complexity, occlusion conditions and simulation visual quality.
>
> Most of the simulated environments in VQ-BeT are either not image-based or not related to the focus of this work which is object manipulation. The environments that involve object manipulation from pixels are FrankKitchen (which we experiment on), and PushT. Notice that VQ-BeT outperforms baselines on goal-conditioned ‘Image PushT’ but obtains relatively poor results (0.1 out of 1, see Table 1 in the [VQ-BeT paper](https://arxiv.org/abs/2403.03181 )). We believe that VQ-BeT completely fails in multi-object manipulation from images because this is a **very hard problem** to solve directly from raw pixel observations without incorporating any prior knowledge of the world in the form of (entity-centric) structure. This is precisely what we aim to demonstrate in our experiments and why we chose to deploy an excellent SOTA algorithm such as VQ-BeT on environments with high multi-object complexity, proving this point.
>
> In the following, we provide details about each environment in VQ-BeT in comparison to the related environment in our work:
>
> *BlockPush and UR3 BlockPush*: These environments are solved in the VQ-BeT paper from **simulation state** and not from images. In addition, the start and goal locations of the cubes are small fixed regions on the table and the goal variability comes from which cube is pushed to which one of the 2 goals. In contrast, the PushCube IsaacGym environment is perceived from images, and start and goal configurations are sampled uniformly at random on the table surface at the beginning of each episode.
>
> *PushT*: The simulated environment introduced in the Diffusion-Policy paper that was also used in VQ-BeT is a 2D environment with a *single* T-block and a point robot, where the goal is visually present on the manipulation surface. This type of goal-specification is less practical than providing a goal image (would require drawing the goal on the table before each episode) and makes it easier for a standard CNN to track progress in reaching the goal based on how much of the goal ‘shadow’ is occluded. In contrast, the PushT IsaacGym environment is 3D, contains multiple T-blocks, has a full 7-DOF robot arm which requires handling occlusion, goals are specified by separate images requiring that the agent learn the relationship between entities across images.
>
> *Multimodal Ant*: This is a locomotion environment where the observation space is the **simulation state** and not images. Our work focuses on object manipulation from pixels, therefore we did not experiment with this environment. It could be interesting to see if an entity-centric structure provides value in learning locomotion tasks from pixels, although we believe the advantage over unstructured image representations in this case would be less apparent.

---

> > ### Author Response · Authors · 2024-11-20
> >
> > *nuScenes*: A self-driving environment where the observation space already assumes access to an object-centric decomposition. Furthermore, it assumes access to object classes and object tracking information (see Figure 11 in the last page of the Appendix of the [VQ-BeT paper](https://arxiv.org/abs/2403.03181 )). In contrast, we do not make these assumptions and propose a method that acquires a representation from images in an unsupervised manner and can leverage it for sequential decision-making without additional assumptions on the structure of the decomposition other than that it is object-centric. Self-driving is an interesting domain to incorporate entity-centric structure but the variations in assumptions and the fact it is not an object manipulation environment (which we clearly convey is the focus of our work) are the main reasons we do not include such experiments.
> >
> > **Why Diffusion Models**
> >
> > A major challenge in behavioral cloning is capturing multi-modal action distributions often present in offline data. This multi-modality is present even in single object manipulation, as often demonstrated in the PushT environment where the agent can maneuver from either side of the T-block in order to push it to the goal (see Figure 1 Left in the [VQ-BeT paper](https://arxiv.org/abs/2403.03181 )). This multimodality increases combinatorially with the number of objects: for example in a task involving 2 T-blocks there are 4 different paths to take that include the order of attending to each object in addition to which side to push each object from.
> >
> > Two main approaches have demonstrated success in capturing such diverse action distributions in the BC setting: (1) Diffusion-based policies (e.g. [Diffuser](https://arxiv.org/abs/2205.09991), [Diffusion Policy](https://arxiv.org/abs/2303.04137v4)). (2) Behavior Transformers ([BeT](https://arxiv.org/abs/2206.11251), VQ-BeT). Diffusion-based policies rely on the stochastic denoising process to capture multiple modes in the data while Behavior Transformers do this with an output tokenization mechanism that uses K-Means (original BeT) or Vector-Quantization (VQ-BeT).
> >
> > Our claim is not that diffusion is better than BeT, but that object-centric structure can greatly benefit such methods when learning from images, and that without this structure, they fail catastrophically when scaling the number of objects. We show this by comparing with both SOTA diffusion-based (Diffuser) and SOTA non-diffusion-based (VQ-BeT) without object-centric structure. We show that a Transformer-based object-centric method that does not account for multi-modality in the demonstrations (EIT+BC) fails as well, showcasing that both object-centric structure *and* multi-modality-capturing models (in our case diffusion models) are necessary for learning.
> >
> > We provide rollouts demonstrating multi-modality in the behaviors generated by EC-Diffuser on our website.
> >
> > **Questions**
> >
> > We hope that the reviewer’s questions have been addressed in the above responses. If not, please point us to any additional points that require clarification.

---

> > > ### Comment · Reviewer_1WDY · 2024-11-27
> > >
> > > Thank you for the detailed reply. It has helped address some of my concerns, and I appreciate the clarification. As a result, I am revising my rating to a 6.

---

> ### Comment · Area_Chair_3EvC · 2024-11-26
> **[ACTION NEEDED] Respond to author rebuttal**
>
> Dear Reviewer,
>
> Now that the authors have posted their rebuttal, please take a moment and check whether your concerns were addressed. At your earliest convenience, please post a response and update your review, at a minimum acknowledging that you have read your rebuttal.
>
> Thank you,
> --Your AC

---

### Official Review · Reviewer_i8yN · 2024-11-04

**Soundness:** 3
**Presentation:** 2
**Contribution:** 3
**Rating:** 6
**Confidence:** 3

**Summary:**

The paper proposes entity-centric diffusion transformer architecture to generate the sequence of actions and latent states conditioned on a trajectory of image observations across multiple views. The paper claims that object centric representations facilitate composite generalization of behavior cloning agents. Experiments are conducted across three tasks namely, PushCube, PushT and FrankaKitchen to show the effectiveness of the proposed method.

**Strengths:**

The proposed method leverages object-centric representations to generate action-state sequences for Behavior Cloning and shows that it generalizes to manipulating multiple objects. Experiments are conducted across three different tasks.

**Weaknesses:**

The novelty of the proposed method is limited as it combines existing methods on entity-centric representation with diffusion policy. It builds upon the existing method [1] to use diffusion models instead of transformers for generating the action-state sequences for object manipulation. Diffusion models have already been shown to be useful for object manipulation tasks in prior works [2, 3].

[1] Haramati et. al, Entity-Centric Reinforcement Learning for Object manipulation from pixels.

[2] Mishra et. al. ReorientDiff: Diffusion Model based Reorientation for Object Manipulation

[3] Chi et. al. Diffusion policy: Visuomotor policy learning via action diffusion.

Missing experiments on evaluating the learned policies on real world robots.

Figure 4 and Figure 5 overlaps with the captions of the Tables above them.

**Questions:**

Is the DLP pretraining done separately for each task in IsaacGym environment?

Further, if PushCube and PushT image observations were used together for pretraining DLP, can it handle the composition where both objects are present together?

Can the method generalize in manipulating novel objects? For example, what is the success rate when T-cubes are added to the Task 1 environment instead of normal cubes.

Since competing methods VQ-BET and EIT-BC work well for single objects, for multiple objects, can they be used sequentially manipulating one object at a given time? What is the performance of this baseline?

What is the performance of the method when using only the frontal view to train the diffusion model? Are two views essential?

Why is classifier-free guidance not used in the diffusion model? Can classifier-free guidance improve the success rate?

---

> ### Author Response · Authors · 2024-11-20
>
> We thank the reviewer for acknowledging our method’s generalization capabilities, and for their in-depth questions. We provide detailed answers below, along with new results from additional experiments.
>
> **Novelty**
>
> To the best of our knowledge, we are the first to use diffusion models on top of unsupervised object-centric image representations for *decision-making*, as well as the first to demonstrate such zero-shot compositional generalization capabilities in the behavioral cloning (BC) setting.
>
> A major challenge in BC is capturing multi-modal action distributions often present in offline data. This multi-modality is present even in single object manipulation, but increases combinatorially with the number of objects. Diffusion models have demonstrated success in capturing diverse action distributions in the BC setting but without entity-centric structure, they fail catastrophically when scaling the number of objects (as our experiments reveal). One aspect of our work’s novelty is identifying the potential of this combination and demonstrating that it can solve challenging multi-object tasks from image observation data in a manner that facilitates zero-shot compositionally generalizing behaviors.
>
> To achieve this, we developed a novel Transformer architecture that includes conditioning on actions by incorporating them as a particle state as well as using Adaptive Layer Normalization throughout the denoising diffusion process. In addition, due to the unique structure of the DLP representation, our design choices include equivariance considerations, such as removing positional embeddings. Finally, we demonstrate SOTA results on challenging simulated robotic manipulation environments compared to strong baselines. We compare with prior works that do not use object-centric representations (VQ-BeT, Diffuser), works that use an object-centric representation but do not generate states (Diffusion Policy) or do not use diffusion (EIT+BC) to showcase the advantages of our specific design choices.
>
> While our work builds on some aspects of [ECRL](https://arxiv.org/abs/2404.01220), applying them in our setting is not trivial. In this work we show that a naive adaptation of ECRL to the behavioral cloning setting is not able to learn multi-object manipulation (see results for EIT+BC baseline in Table 1), demonstrating that our approach of using diffusion models and our novel architecture accounts for the distinct challenges that arise when learning from offline demonstration data containing diverse behaviors.
>
> **Real Robots**
>
> We kindly refer the reviewer to the general response regarding real-world applicability.
>
> **Questions**
>
> *Is the DLP pretraining done separately for each task in IsaacGym environment?*
>
> DLP pre-training is done separately for each environment (e.g., PushCube or PushT) but not for each task (i.e. different number of objects). For the IsaacGym environments we train a total of 2 DLP models, one for PushCube and one for PushT, each trained on 2 views of an environment with 6 and 3 objects respectively. DLP generalizes to images with a different number of objects than in training. We have added additional details about DLP pretraining in Appendix B under each environment in the new revision.
>
> *Further, if PushCube and PushT image observations were used together for pretraining DLP, can it handle the composition where both objects are present together?*
>
> If we understand the reviewer correctly, the question is whether DLP has compositional generalization capabilities. DLP is a *local* representation that encodes scenes as sets of entities, with each entity representing a small region of the image. During training, DLP learns to encode distinct objects from various scenes in its latent space. At inference time, it can compose these objects into new configurations, even if they were not seen together during training.
> Our generalization results (see Figure 6) demonstrate that DLP can handle scenes with a different number of objects than it was trained on, which requires composing objects it encountered individually during training. This is analogous to the scenario described in the question. Based on this evidence, we expect that DLP would successfully handle scenes containing both T-blocks and cubes, even if it was trained exclusively on images containing only one of these objects at a time.

---

> > ### Author Response · Authors · 2024-11-20
> >
> > *Can the method generalize in manipulating novel objects? For example, what is the success rate when T-cubes are added to the Task 1 environment instead of normal cubes?*
> >
> > We evaluate our policy on PushCube with novel objects including star-shaped objects, rectangular cuboids, T-shaped blocks and cubes with unseen colors. The results can be found in Appendix D.3, Table 14. We see that EC-Diffuser coupled with DLP is able to generalize zero-shot with little to no drop in performance to new colors as well as new shapes (star, rectangular cuboid). When replacing cubes with T-shaped blocks there is a significant drop in performance although success rate is better than random, suggesting some zero-shot generalization capabilities in this case as well. We see that our policy handles new objects well in cases where they behave similarly in terms of physical dynamics and less when they are significantly different, which is expected. We believe the zero-shot results on T-blocks indicate that our policy can be adapted to new objects with relatively little additional data.
> >
> > *Since competing methods VQ-BET and EIT-BC work well for single objects, for multiple objects, can they be used sequentially manipulating one object at a given time? What is the performance of this baseline?*
> >
> > This is an interesting question and is one that we have thought about deeply, which is also a subject of previous work in the object-centric decision-making literature.
> >
> > The short answer to this question is twofold: (1) Devising a mechanism for decomposing the multi-object task such that a single-object policy could be deployed to solve for single objects sequentially is far from trivial, and usually requires incorporating a high-level policy (to select the current task/object) and a low-level policy (to solve the single-object task). (2) Sequentially solving for single objects assumes independence between single-object tasks, which is not always the case.
> >
> > In the following, we elaborate on these points:
> > [SMORL](https://arxiv.org/abs/2011.14381) proposes an object-centric RL algorithm that does exactly what you suggest: learning single-object manipulation during training and using a simple meta-policy during inference that cycles between objects sequentially until all objects’ goals are reached. A major underlying assumption in this approach is that the multi-object task is completely decomposable, i.e. there is no dependency between single-object tasks/goals. This is not always the case. This is arguably never precisely the case in realistic manipulation environments due to physical interaction between objects. Another important note is that although SMORL trains on single-object manipulation, it still requires learning in a multi-object environment in order to solve multi-object tasks due to their training-to-inference procedure. This requirement stems from the fact that the agent needs to learn to ignore objects that are present in the environment but do not correspond to a chosen single-object goal.
> > [SRICS](https://arxiv.org/abs/2109.04150) proposes an algorithm that can handle inter-object dependencies. [ECRL](https://arxiv.org/abs/2404.01220) presents a suite of environments that more clearly requires handling object-object interaction and demonstrates that SMORL fails while their method does not since it considers all objects in the environment in every decision step.
> >
> > Now let us assume the task is approximately decomposable. The main challenge in this case is how to efficiently decompose a multi-object task to single-object tasks. One cannot simply apply a single-object policy to a multi-object environment zero-shot since the input observations and goals would be out-of-distribution. In addition, the single-object policy has never encountered the need to reason between multiple objects, e.g. deciding on a manipulation order, and thus it cannot be expected to generalize in that respect. Therefore, a more sophisticated mechanism is required in order to solve a multi-object task with a single-object policy. We do not know of an algorithm that does this, and cannot think of a simple implementation that does not require training with multiple objects and several manual interventions during training and inference. We believe developing such an algorithm is an interesting direction for future research.

---

> > > ### Author Response · Authors · 2024-11-20
> > >
> > > *What is the performance of the method when using only the frontal view to train the diffusion model? Are two views essential?*
> > >
> > > We provide results ablating the contribution of multiple viewpoints in the IsaacGym environments in Appendix D.3, Table 12. These results show the significance of multiple views for solving these tasks. We believe the advantage in using multiview perception stems from the partial observability of the environment, and contributes in two aspects: (1) Multiview helps handling occlusions. (2) A second view provides complimenting observational information that aids the model with internally inferring key attributes such as ground-truth entity position and orientation. Our experimental results are consistent with a similar ablation in the ECRL paper.
> > >
> > > *Why is classifier-free guidance not used in the diffusion model? Can classifier-free guidance improve the success rate?*
> > >
> > > If the reviewer refers to classifier-free guidance as a mechanism involving a separate, potentially inferior model to guide samples toward high-likelihood regions, we agree that this could potentially improve performance. However, implementing such guidance requires additional effort to train a separate guiding model for each policy. Moreover, it necessitates running two networks during inference, which increases the computational cost and inference time for action generation. Additionally, for a fair comparison with prior diffusion-based BC methods, we chose not to deviate from the original diffusion process.
> > >
> > > Alternatively, if the reviewer is referring to guidance based on a conditioning signal, it is unclear to us how such guided sampling would contribute in the behavioral cloning context or what form of guidance would be relevant. Could the reviewer clarify what guidance signal they propose and why they believe it would enhance performance in this setting?
> > >
> > > A common guidance signal in RL is derived from rewards or value functions. However, our setting assumes access to expert demonstrations (i.e., behavioral cloning) and does not include reward-labeled data. Extending our method to offline RL, where rewards are available and could potentially serve as a guidance signal, is an interesting direction for future work.

---

> ### Comment · Reviewer_i8yN · 2024-11-24
>
> Thank you to the authors for the detailed rebuttal that addressed many of my concerns.
>
> Regarding the DLP pretraining, my question was whether training the DLP model with image data containing both cubes and T-blocks would improve the success rate—essentially, whether DLP's object-centric representations benefit from additional data. The results in Table 14 demonstrate some generalization to other shapes, which partially addresses my concern. However, for entirely novel objects like T-blocks, the performance drops, so an analysis on whether better object-centric representations could enhance these results will be useful.
>
> My other question pertains to classifier-free guidance, not classifier guidance, which I understand requires training a separate model and increases computational demands. With classifier-free guidance, null-conditioning (e.g., using zero vectors for current state) is applied during training, and a guidance scale hyperparameter is used during inference. Typically, such guidance  in image generation improves the quality while reducing the diversity. I wanted to understand whether this approach helps in this setting.

---

> > ### Author Response · Authors · 2024-11-25
> >
> > We thank the reviewer for their response and further engagement.
> >
> > **DLP Pretraining**
> >
> > Generalization to entirely novel objects is not expected when their dynamics are significantly different (e.g. T-blocks when trained on cubes), because this is essentially an entirely new task. Even if the DLP representation was trained on data containing T-blocks for example, the diffusion model did not train on generating state-action trajectories with T-blocks. Therefore, we do not expect this to improve performance. On the contrary, we believe that performance will be worse in this case because DLP would produce latent particles that are out-of-distribution for EC-Diffuser.
> >
> > We have added a figure of the decoded DLP representations of an environment with T-blocks and the generated EC-Diffuser trajectories, where both DLP and EC-Diffuser were trained on data containing only cubes (see Figure 9, Appendix). We believe this would be of interest to the reviewer and might strengthen our point in the previous paragraph. In this figure we can see that the DLP represents the T-block as a composition of cubes, composing the overall scene from the objects it is trained on. This can be seen as a form of compositional generalization. While we find this is an interesting capability on its own, this generalization is only *visual* and does not translate to better action or future state generation. EC-Diffuser is still trained for the dynamics of individual cubes and cannot account for dynamics of cubes that are “mended” together to form a T-block.
> >
> > **Classifier-free Guidance**
> >
> > We thank the reviewer for the clarification regarding diffusion guidance. Our initial understanding was that the reviewer referred to the following common approach: in order to push the generated samples into high-likelihood regions, the model should be conditioned on corrupted samples generated from an inferior model (that is also trained), as become popular in recent image/video generation approaches (for example, see https://arxiv.org/abs/2411.10257).
> > As for the suggested mechanism proposed by the reviewer, we posit that any method that improves the generated samples quality of diffusion models, may be utilized in our setting to improve the performance. For a fair comparison with the diffusion-for-decision-making literature, we maintain the original setting and don’t use any guidance signal, but we agree that it is an interesting research direction which is orthogonal to our contributions. If the reviewer still thinks it could strengthen our paper and can direct us to a specific guidance mechanism implementation they have in mind, we are happy to experiment with it and include the results in the next revision.

---

> > > ### Author Response · Authors · 2024-12-03
> > >
> > > Dear reviewer, based on our discussion, we have the impression that most of your major concerns about novelty and generalization have been addressed. We have presented additional experiments including zero-shot generalization of EC-Diffuser and DLP to novel objects, single-view ablation and real-world LanguageTable. We hope that our preliminary results on LanguageTable have helped address your concerns about testing our approach on real-world robots.

---

### Author Response · Authors · 2024-11-20
**General Response**

We sincerely appreciate the reviewers' time and effort invested in providing us constructive and actionable feedback. Following is a general response regarding several common points that were raised by the reviewers. In addition, we have uploaded a new revision incorporating content that addresses the reviewers’ questions and suggestions. We summarize the additions at the end of this response. **Modifications are highlighted in blue in the new revision for the reviewer’s convenience**.

**Choice of Object-centric Representation**

Our work advocates the use of unsupervised object-centric image representations in order to facilitate entity-level reasoning. The simplest definition for a *good* object-centric representation for our purposes is one that effectively facilitates downstream decision-making. As our experimental results demonstrate, DLP not only fulfills this criterion but also enables zero-shot generalization in certain scenarios.

Following some of the reviewers' questions and requests, we conducted additional experiments with an alternative object-centric representation: Slot Attention. On FrankaKitchen, EC-Diffuser along with Slot Attention, achieved a new SOTA (see Appendix D.2, Table 13 in the new revision), while in PushCube performance was significantly lower than using DLP but still better than non-object-centric baselines.
In the following points we elaborate on why we believe DLP is better suited than slot-based models for the types of tasks we address:
- The latent particles are composed of explicit attributes, such as position and scale in pixel space, which are learned entirely unsupervised. This provides a valuable inductive bias for object manipulation tasks, as it offers a direct and interpretable signal for the locations of objects, joints, and other areas of interest in the scene.

- The latent particles represent small, localized regions of the image (also referred to as glimpses), unlike slots that encode information spanning the entire image. This makes particles a more lightweight latent representation, reducing the computational cost of diffusion. Additionally, the ability to allocate a larger number of particles per image (~24 compared to the typical ~7 slots) enhances the representation of distinct objects and supports compositional generalization to scenes with more objects than seen in training.

- Extracting particles during inference is simple and fast since the encoder is a feed-forward network, as opposed to extracting slots which is an iterative procedure.

**Real World Applicability**

While we do not experiment with real robots in the physical world, we believe that our general approach, as well as our proposed algorithm, are applicable to real-world object manipulation. We do not see a fundamental limitation in solving real world tasks similar to the ones in our simulated suite using our method. We demonstrate our method’s performance on environments implemented in the high-fidelity IsaacGym simulator, which are more challenging than simulated manipulation environments presented in previous work (see response to reviewer 1WDY regarding comparison to simulated environments used in VQ-BeT). We additionally provide DLP decompositions of real images from the [Language-Table](https://interactive-language.github.io/) dataset (an environment similar to PushCube) in Figure12 (Appendix) of the new revision.
That being said, applying our method in the real world would require enhancement to some aspects of our implementation such as accelerating diffusion sampling for real-time inference, which is not unique to our approach but to all diffusion-based decision-making. We discuss this and more in Section 6 of the paper. We have added a discussion explicitly addressing real-world applicability in the “Limitations and Future Work” portion of this section in the new revision.

**Summarization of Additions**

- Experiments and discussion on using alternative object-centric representations with EC-Diffuser, including Slot Attention (Appendix D.2).
- Experiments and discussion on generalization to novel objects (Appendix D.3).
- Modified Conclusion (Section 6) to incorporate a broader discussion about real-world applicability.
- Details about DLP pretraining in each environment (Appendix B).
- DLP decompositions of real images from the [Language-Table](https://interactive-language.github.io/) dataset (Appendix D.4, Figure 12).
- Rollouts of generalization to novel objects (on the website).
- Rollouts demonstrating multi-modal behaviors (on the website).
- Fixed minor formatting issues pointed out by reviewers.

---

### Author Response · Authors · 2024-11-27
**General Response - An Update on Language-Table Experiments**

We would like to inform the reviewers of additional preliminary experimental results on applying EC-Diffuser on a real world dataset, Language-Table, for which we have included a new section in Appendix D.3. In summary, EC-Diffuser is able to generate high-quality rollouts in the form of particle states from which we visualize the decoded images in Figure 8. For a quantitative evaluation, we measure the prediction l1 loss for the first action (action to be executed). EC-Diffuser achieves a loss of 7e-3 for actions in the validation set and 6e-3 for actions in the training set (for reference).

We believe that this, in addition to the DLP reconstructions that we have already provided, shows promise for the application of EC-Diffuser in real-world scenarios.

We thank the reviewers for their responsiveness and engagement throughout the rebuttal period thus far.

---

### Meta-Review · Area_Chair_3EvC · 2024-12-20

**Metareview:**

This paper introduces a framework for behavioral cloning using an object-centric architecture. The architecture is based on Deep Latent Particles (DLP) and a transformer module, optimised using a diffusion objective.

Reviewers agree that the paper is well-written and contains a solid experimental evaluation (with reasonable baselines and ablations).

One major concern raised by reviewers is that this paper's strength is mostly in its engineering advancements instead of providing a truly novel approach or method. This is a valid concern, yet the overall quality of the paper, results, and combination of techniques makes this an interesting result worth highlighting in the community.

**Additional Comments On Reviewer Discussion:**

Reviewer consensus did not change during discussion.

---

### Decision · Program_Chairs · 2025-01-22

Accept (Poster)